# Symmetry in Neural Network Parameter Spaces

**Bo Zhao**                                                         *bozhao@ucsd.edu*
*University of California, San Diego*

**Robin Walters**                                          *r.walters@northeastern.edu*
*Northeastern University*

**Rose Yu**                                                         *roseyu@ucsd.edu*
*University of California, San Diego*

**Reviewed on OpenReview:** *https://openreview.net/forum?id=jLpWq5QY6I*

## Abstract

Modern deep learning models are highly overparameterized, resulting in large sets of parameter configurations that yield the same outputs. A significant portion of this redundancy is explained by symmetries in the parameter space—transformations that leave the network function unchanged. These symmetries shape the loss landscape and constrain learning dynamics, offering a new lens for understanding optimization, generalization, and model complexity that complements existing theory of deep learning. This survey provides an overview of parameter space symmetry. We summarize existing literature, uncover connections between symmetry and learning theory, and identify gaps and opportunities in this emerging field.

## 1 Introduction

Despite their remarkable empirical success, neural networks remain not fully understood from a theoretical standpoint. Designing effective architectures often relies on heuristics, and training outcomes can be highly sensitive to initialization and optimization details. The rapid growth of model sizes—exemplified by the recent explosion in the size of language and other generative models (Kaplan et al., 2020)—further complicates efforts to characterize training dynamics and generalization behavior. While theoretical work has clarified aspects of overparameterization (Belkin et al., 2019; Neyshabur et al., 2019) and implicit regularization in gradient descent (Soudry et al., 2018), we still lack a unified understanding of how the structure of parameter space shapes the loss landscape and affects optimization.

This survey proposes parameter space symmetry as a valuable, underexplored perspective for understanding neural networks. Parameter space symmetries are transformations of neural network parameters that leave the output unchanged, e.g. permuting neurons in a hidden layer (Figure 1). One of the most important consequences of these symmetries is that they induces nontrivial structure in the level sets of the loss function. In overparameterized networks, where many parameter configurations can yield the same function output, the loss landscape often contains high dimensional manifolds of global minima (Cooper, 2018). Symmetry accounts for much of this degeneracy by mapping parameters within a level set without changing the network's function. Understanding these equivalence classes is essential for analyzing generalization, optimization dynamics, and the loss landscape in deep learning.

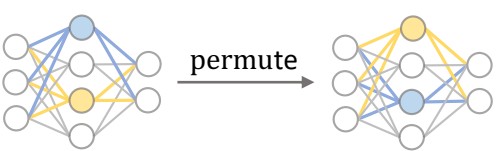

Figure 1: Example of parameter space symmetry from neuron permutation. Swapping hidden units and their weights yields a different parameter configuration with the same function.

While symmetry in the data space has been central to geometric deep learning and equivariant models (Bronstein et al., 2021), symmetry in the parameter space has only recently begun to receive sustained attention. Recent work explores parameter symmetry in diverse contexts, including loss landscapes and mode connectivity (Garipov et al., 2018; Draxler et al., 2018), conserved quantities and training dynamics (Kunin et al., 2021; Tanaka & Kunin, 2021), symmetry-based optimization and model averaging (Entezari et al., 2022; Ainsworth et al., 2023), and symmetry-aware sampling in Bayesian inference (Wiese et al., 2023). This survey aims to define symmetries in neural network parameter spaces, unify existing theoretical and algorithmic perspectives, and highlight their relevance to learning theory, optimization, and model analysis. We hope that a clearer understanding of parameter space symmetries will lead to more principled approaches in deep learning theory and practice.

The rest of this paper is organized as follows. Section 2 reviews definitions and examples of symmetries in the parameter space of neural networks. Section 3-5 collectively survey the theoretical and algorithmic applications of parameter space symmetry in deep learning: Section 3 discusses the role of symmetry in the geometry and topology of loss level sets, with a focus on minima; Section 4 lists progress in optimization by incorporating the knowledge of parameter space symmetry; Section 5 discusses conserved quantities in gradient flow associated with parameter space symmetry and their applications in understanding learning dynamics. Section 6 then discusses how parameter space symmetry interacts with data and representation symmetries, as well as applications in tasks where neural network parameters are treated as data. We conclude with a list of open questions and opportunities in Section 7.

## 2   Parameter Space Symmetry

In this section, we explore various definitions and examples of parameter space symmetry, starting with transformations of the parameters that preserve the feedforward function. Then we expand our discussion to include relaxed definitions of symmetry, which preserve the overall loss function or the feedforward function's value on subsets of data. Along with these definitions, we provide a list of known symmetries and show how they arise from the architecture of neural networks. Finally, we discuss the impact of symmetries on parameter identifiability and whether the known symmetries are complete.

### 2.1   Loss-Invariant Parameter Transformations and Symmetry

Let $\Theta$ be the space of parameters and $\mathcal{D}$ be the space of data. In supervised learning, $\mathcal{D}$ is decomposed into $\mathcal{D}_{\text{input}} \times \mathcal{D}_{\text{target}}$, where $\mathcal{D}_{\text{input}}$ and $\mathcal{D}_{\text{target}}$ correspond to the space of inputs and target labels, respectively. Let $f \colon \Theta \times \mathcal{D}_{\text{input}} \to \mathcal{D}_{\text{target}}$ be a neural network function, and $c \colon \mathcal{D}_{\text{target}} \times \mathcal{D}_{\text{target}} \to \mathbb{R}$ be a function that measures the discrepancy between the neural network's prediction and the ground truth label. The loss function $L \colon \Theta \times \mathcal{D} \to \mathbb{R}$ is the composition of $f$ and $c$. That is, for $(\theta, (x, y)) \in \Theta \times \mathcal{D}$, we define the loss $L(\theta, (x, y)) := c(f(\theta, x), y)$. With occasional exceptions (Bassey et al., 2021; Huang et al., 2018), the parameter space of a neural network is typically a real vector space.

Symmetries are transformations that preserve certain properties of an object. In this paper, we focus on parameter space symmetry, which are transformations of a neural network's parameters that preserve loss. Mathematically, this is the set of bijective transformations $T \colon \Theta \to \Theta$ such that $L(w) = L(T(w))$. One may consider restricted sets of symmetry by imposing additional constraints such as linearity or smoothness.

The set of all such transformations forms a group under function composition. To see this, we first examine its structure and properties. Viewing the transformations as functions, we are able to define the composition of two transformations. The composition of two loss-invariant transformations results in another loss-invariant transformation. Since function compositions are associative, the composition of these transformations are associative. Additionally, the identity transformation is always loss preserving, therefore included in our set. Finally, since each transformation is bijective, an inverse exists for every element in the set. These properties of the set of symmetry transformations under composition satisfy the definition of a group.

**Definition 2.1** (Group). *A group is a set $G$ together with a composition law $\circ$ that satisfies (1) Associativity: $(g_1 \circ g_2) \circ g_3 = g_1 \circ (g_2 \circ g_3)$ for all $g_1, g_2, g_3$ in $G$; (2) Identity: There exists an identity element $e$ in $G$ such*

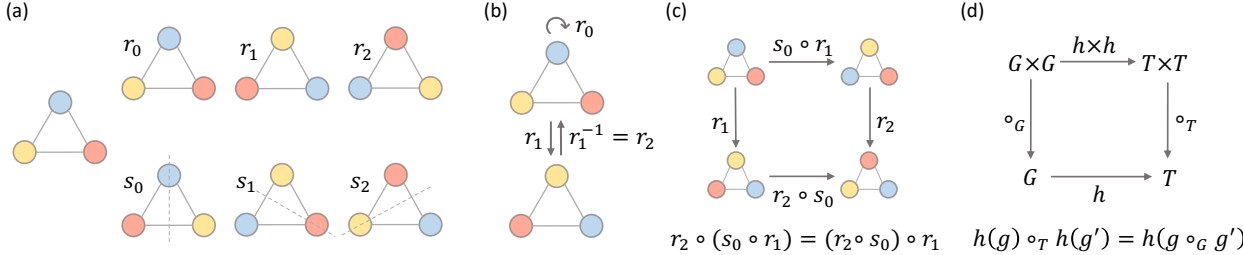

Figure 2: An example of symmetry and group actions. (a) The dihedral group $D_3$, which is the group of symmetries of a regular triangle. $D_3$ has six elements: three rotations $r_0, r_1, r_2$ and three reflections $s_0, s_1, s_2$. (b) The identity element $r_0$ does not change the object. Each group element has an inverse. (c) Composition of transformations are associative. (d) A group action can be defined as a homomorphism $h$ from a group $G$ to the group of symmetry transformations $T$.

that $g \circ e = g$ and $e \circ g = g$ for all $g$ in $G$; (3) Inverse: For every element $g$ in $G$, there exists an inverse element $g^{-1}$ such that $g \circ g^{-1} = e$ and $g^{-1} \circ g = e$.

**Example 2.1.** *We illustrate a group of symmetry transformations on the dihedral group $D_3$, which represents symmetries of regular triangles (Figure 2). $D_3$ has six elements: three rotations $0°, 120°, 240°$ ($r_0, r_1, r_2$) and three reflections across axes passing through each vertex and the midpoint of the opposite side ($s_0, s_1, s_2$). The composition of these transformations is also a symmetry. The identity element $r_0$ does not change the triangle. Each transformation has an inverse, which is also in the group. Composition of transformations are associative.*

Following Definition 2.1, the set of all loss-invariant and bijective transformations, which we denote by $G_{\Theta, L}$, forms a group. In practice, we typically consider specific subgroups of $G_{\Theta, L}$, since the full group is often difficult to characterize.

For generality and to simplify analysis, we describe symmetry using abstract groups, which focus on the algebraic properties of groups detached from specific transformations. An example that will appear frequently is the $n \times n$ general linear group over $\mathbb{R}$. This group, denoted by $\mathrm{GL}_n(\mathbb{R})$, consists of invertible $n \times n$ real matrices, with composition defined by matrix multiplication. We will also encounter several subgroups of $\mathrm{GL}_n(\mathbb{R})$, including the orthogonal group $O_n(\mathbb{R})$ which consists of real matrices whose transpose equal inverse, and the positive scaling group $\mathbb{R}_{>0}^h$ which consists of diagonal matrices with positive diagonal entries. Another group relevant to neural network parameter spaces is the symmetric group $S_n$, which consists of permutations of the set $\{1, 2, ..., n\}$.

Group actions connect the abstract concept of groups with concrete sets of transformations. A group action is a structure-preserving map from a group into a group of transformations (Figure 2d).

**Definition 2.2** (Group action). *An action of a group $G$ on a set $S$ is a map $\cdot: G \times S \to S$ that satisfies $e \cdot s = s$ for all $s \in S$ and $g \cdot (g' \cdot s) = (gg') \cdot s$ for all $g, g'$ in $G$ and all $s$ in $S$.*

A parameter space symmetry can then be described as a group action on the space of parameters that leaves the loss unchanged. In many machine learning settings, these group actions are linear. This leads to the concept of a representation, which maps group elements to invertible matrices and enables the group to act on a vector space by linear transformations.

**Definition 2.3** (Representation). *A representation of a group $G$ is a homomorphism $\rho: G \to \mathrm{GL}_n(\mathbb{R})$, meaning that $\rho(g_1 g_2) = \rho(g_1)\rho(g_2)$ for all $g_1, g_2 \in G$.*

## 2.2 Functional Neural Network Symmetry

Parameter space symmetry can be defined in various ways, depending on the transformation preserved, i.e. the neural network function or the loss function, and the scope of the data considered, which can range from all possible data, subsets of data, or a particular distribution (Figure 5).

This section focuses on the strictest form of parameter space symmetry, which involves transformations that leave the neural network output invariant across all data. Such symmetry preserves the feedforward function. We discuss various general definitions in Section 2.3.

**Definition 2.4** (Functional neural network symmetry). *Let $\Theta$ be the parameter space of a neural network $f\colon \Theta \times \mathcal{D}_{input} \to \mathcal{D}_{target}$. A parameter space symmetry of $f$ is a (possibly nonlinear) action of a group $G$ on $\Theta$ that leaves $f$ invariant,*

$$f(g \cdot \theta, x) = f(\theta, x), \quad \forall g \in G, \quad \forall \theta \in \Theta, \quad \forall x \in \mathcal{D}_{input}.$$

*The group $G$ is called a symmetry group of $f$.*

### 2.2.1 Examples: Symmetries in Common Neural Network Components

Different neural network model architectures give rise to different parameter space symmetries. In the following examples, we examine common components of neural networks and identify specific symmetries associated with each. We begin with a linear network, which offers a clean setting to illustrate how parameter symmetries emerge from rescaling adjacent layers. We work towards realistic examples afterwards.

**Example 2.2** (Linear). *Consider a two-layer linear neural network $f_{linear}(W_2, W_1, b_2, b_1, X) = W_2(W_1X + b_1) + b_2$, with $(W_2, W_1, b_2, b_1) \in \Theta = \mathbb{R}^{m \times h} \times \mathbb{R}^{h \times n} \times \mathbb{R}^m \times \mathbb{R}^h$ and $X \in \mathbb{R}^{n \times k}$. This architecture has a $GL_h(\mathbb{R})$ symmetry, acting on $\Theta$ by $g \cdot (W_2, W_1, b_2, b_1) = (W_2 g^{-1}, gW_1, b_2, gb_1)$, for $g \in GL_h(\mathbb{R})$ since*

$$\begin{aligned}
f_{linear}(g \cdot (W_2, W_1, b_2, b_1), X) &= W_2 g^{-1}(gW_1X + gb_1) + b_2 \\
&= W_2(W_1X + b_1) + b_2 \\
&= f_{linear}(W_2, W_1, b_2, b_1, X).
\end{aligned}$$

Symmetries of similar forms appear in networks with activation functions, which are more commonly used than linear networks. In particular, many common activation functions are equivariant under a nontrivial group, which leads to following symmetries (Figure 3b).

**Proposition 2.5** (Zhao et al. (2023)). *Let $\sigma\colon \mathbb{R}^h \to \mathbb{R}^h$ be a function that satisfies $\sigma(gZ) = \rho(g)\sigma(Z)$ for a group $G$ and a representation $\rho\colon G \to \mathrm{GL}_h(\mathbb{R})$. Consider a function $f\colon \Theta \times \mathcal{D} \to \mathbb{R}^{m \times n}$, $(W_2, W_1, b_2, b_1, X) \mapsto W_2\sigma(W_1X + b_1) + b_2$, where $(W_2, W_1, b_2, b_1) \in \Theta = \mathbb{R}^{m \times h} \times \mathbb{R}^{h \times n} \times \mathbb{R}^m \times \mathbb{R}^n$ and $X \in \mathbb{R}^{n \times k}$. Then, $f$ admits a functional parameter space symmetry defined by $g \cdot (W_2, W_1, b_2, b_1) \mapsto (W_2\rho(g^{-1}), gW_1, b_2, gb_1)$.*

The symmetries in the next three examples result from equivariance of various pointwise activation functions. A function $\sigma\colon \mathbb{R}^h \to \mathbb{R}^h$ is called pointwise if it is defined as $\sigma(z)_i = \sigma_i(z_i)$ for some scalar functions $(\sigma_i\colon \mathbb{R} \to \mathbb{R})_{i=1}^h$. In practice, $\sigma_i$ is usually the same across all indices $i$. A pointwise function $\sigma\colon \mathbb{R}^h \to \mathbb{R}^h$ is homogeneous if there exists a degree $\alpha \in \mathbb{R}_{>0}^h$ such that $\sigma(cz)_i = c^{\alpha_i}\sigma(z)_i$ for all $c \in \mathbb{R}_{>0}$ and $z \in \mathbb{R}^h$. Common homogeneous functions include ReLU, LeakyReLU, and monomials. Since the bias terms are transformed similarly as the other weights, we omit the bias terms in the following examples for brevity.

**Example 2.3** (Homogeneous Activation). *Consider a two-layer neural network $f(W_2, W_1, X) = W_2\sigma(W_1X)$ with a homogeneous function $\sigma$ of degree $\alpha$, where $(W_2, W_1) \in \Theta = \mathbb{R}^{m \times h} \times \mathbb{R}^{h \times n}$ and $X \in \mathbb{R}^{n \times k}$. A symmetry group of this architecture is the positive scaling group $\mathbb{R}_{>0}^h$, which consists of diagonal matrix with positive diagonal entries and act on $\Theta$ by $g \cdot (W_2, W_1) = (W_2 g^{-\alpha}, gW_1)$, for $g \in \mathbb{R}_{>0}^h$ (Badrinarayanan et al., 2015).*

**Example 2.4** (Tanh). *Consider a two-layer hyperbolic tangent neural network $f_{tanh}(W_2, W_1, X) = W_2\tanh(W_1X)$, where $(W_2, W_1) \in \Theta = \mathbb{R}^{m \times h} \times \mathbb{R}^{h \times n}$ and $X \in \mathbb{R}^{n \times k}$. This architecture has a sign-flip symmetry, $\mathbb{Z}_2^n$, that consists of diagonal matrix with all diagonal entries in $\{1, -1\}$, acting on $\Theta$ by $g \cdot (W_2, W_1) = (W_2 g^{-1}, gW_1)$ (Chen et al., 1993).*

**Example 2.5** (Radial neural network (Ganev et al., 2022)). *Often appearing in equivariant neural networks (Weiler et al., 2018; Weiler & Cesa, 2019), a radial rescaling activation $\sigma\colon \mathbb{R}^h \to \mathbb{R}^h$ has the form $\sigma(z) = f(\|z\|)z$ for some function $f\colon \mathbb{R} \to \mathbb{R}$. A two-layer neural network $f(W_2, W_1, X) = W_2\sigma(W_1X)$ with a radial rescaling activation $\sigma$ has an $O_h(\mathbb{R})$ symmetry, acting on $\Theta$ by $g \cdot (W_2, W_1) = (W_2 g^{-1}, gW_1)$, for $g \in O_h(\mathbb{R})$.*

When $\sigma$ is $G$-invariant, which means $\sigma(gZ) = \sigma(Z)$, there exists a group action that acts on only the input weights of $\sigma$ (Figure 3a). We illustrate this in the next two examples.

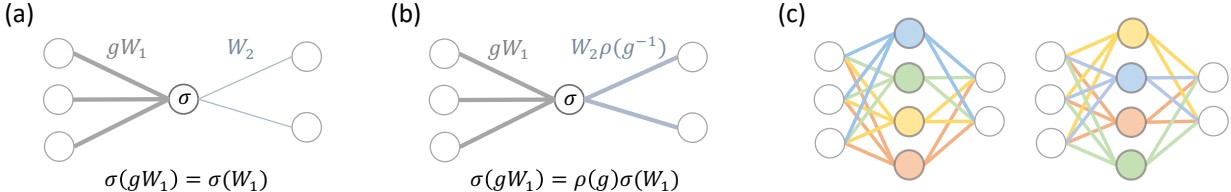

Figure 3: Parameter space symmetries arising from neural network architectures. Each circle in the computation graphs represents a neuron or a component in the architecture. Each line segment represents one or a set of parameters. (a) Scaling of the incoming weights of an invariant activation (Examples 2.6, 2.7). (b) Scaling of the incoming and outgoing weights of an equivariant activation (Examples 2.2-2.5). (c) Permutation of neurons (Example 2.8) or components (Examples 2.9) together with their associated weights.

**Example 2.6** (Batch Normalization (Ioffe & Szegedy, 2015)). *Batchnorm standardizes inputs of a layer across a mini-batch: $BN(Z) = \frac{Z - E[Z]}{\sqrt{Var[Z]}}$, where the expectation and variance are computed row-wise over $Z$. Assuming $Z$ is a linear feature $Z = WX$ where $W \in \Theta = \mathbb{R}^{m \times n}$ and $X \in \mathbb{R}^{n \times k}$, then the batchnorm function has a positive scaling symmetry. Batchnorm is equivariant with respect to the group $\mathbb{R}^m_{>0}$, which consists of diagonal matrix with positive diagonal entries, acting on $\Theta$ by $g \cdot W = gW$, for $g \in \mathbb{R}^m_{>0}$ (Kunin et al., 2021). Other normalization layers, such as layer normalization (Ba et al., 2016), group normalization (Wu & He, 2018), and weight normalization (Salimans & Kingma, 2016), often have scaling symmetry of similar forms.*

**Example 2.7** (Softmax (Bridle, 1989)). *The softmax function $\sigma \colon \mathbb{R}^h \to \mathbb{R}^h$ is defined as $\sigma(z)_i = e^{z_i} / \sum_j e^{z_j}$. Consider the function $f(W, X) = \sigma(WX)$, where $W \in \Theta = \mathbb{R}^{m \times n}$ and $X \in \mathbb{R}^{n \times k}$, and $\sigma$ applies on the rows of $WX$. Then $f$ has a translation symmetry, acting on $\Theta$ by $(g \cdot W)_i = W_i + g$ for each row $W_i$, with $g \in (\mathbb{R}^n, +)$, the additive group over $\mathbb{R}^n$ (Kunin et al., 2021).*

Many architectures have parameter symmetries given by a symmetric group. This symmetry can often be interpreted as permutation of neurons or components along with their associated weights, as shown in Examples 2.7, 2.8 and Figure 3(c). Similar forms of symmetry have also been identified in recurrent neural networks (Albertini et al., 1995).

**Example 2.8** (Pointwise Activation). *Feedforward networks with pointwise activations that use the same scalar function across coordinates have permutation symmetries. Consider a two-layer neural network $f(W_2, W_1, X) = W_2 \sigma(W_1 X)$ with $(W_2, W_1) \in \Theta = \mathbb{R}^{m \times h} \times \mathbb{R}^{h \times n}$, $X \in \mathbb{R}^{n \times k}$, and any pointwise activation function $\sigma$ with $\sigma_i = \sigma_j$ for all $i, j$. There is a permutation symmetry acting on $\Theta$ by $g \cdot (W_2, W_1) = (W_2 g^{-1}, g W_1)$, for $g \in S_h$ (Hecht-Nielsen, 1990).*

**Example 2.9** (Radial basis function networks (Broomhead & Lowe, 1988)). *In a radial basis function (RBF) network $\sum_{i=1}^k w_i \varphi \left( \frac{\|\mathbf{x} - \mathbf{c_i}\|}{b_i} \right)$ with radial function $\varphi \colon \mathbb{R}_+ \to \mathbb{R}$ and parameters $(w_i, b_i, c_i)_{i=1}^k \in \Theta = (\mathbb{R} \times \mathbb{R}_+ \times \mathbb{R}^n)^k$, there is a permutation symmetry acting on $\Theta$ by $\pi \cdot ((w_i, b_i, c_i)_{i=1}^k) = (w_{\pi^{-1}(i)}, b_{\pi^{-1}(i)}, c_{\pi^{-1}(i)})_{i=1}^k$, for $\pi \in S_k$ (Kůrková & Neruda, 1994).*

Viewed as a computational graph, a neural network inherits the symmetries of all its subnetworks (Zhao et al., 2024a). Examples 2.2–2.9 therefore naturally extend to multi-layer networks by applying symmetry to any subset of adjacent layer pairs. Additionally, modern architectures are often constructed from smaller components, many of which have the same form as these examples. Consequently, they admit the same symmetries, which act on a subspace of their parameter space.

### 2.2.2 Example: Symmetries in Transformers

To illustrate how symmetries emerge and combine in popular architectures, we delineate the symmetries in transformers by examining their components (Zhang et al., 2025). Figure 4 visualizes the source of these symmetries.

Table 1: Symmetry in common neural network components (Examples 2.2–2.9). Since larger networks inherit symmetries from their subnetworks, these examples naturally extend to multi-layer networks by applying symmetry to pairs of adjacent layers.

| Name | Architecture | Symmetry Group | Group action |
|---|---|---|---|
| Linear | $W_2 W_1 X$ | $\mathrm{GL}_h(\mathbb{R})$ | $g \cdot (W_2, W_1) = (W_2 g^{-1}, g W_1)$ |
| Homogeneous | $W_2 \sigma_{hom}(W_1 X)$ | $\mathbb{R}_{>0}^h$ | $g \cdot (W_2, W_1) = (W_2 g^{-\alpha}, g W_1)$ |
| Tanh | $W_2 \sigma_{\tanh}(W_1 X)$ | $\mathbb{Z}_2^n$ | $g \cdot (W_2, W_1) = (W_2 g^{-1}, g W_1)$ |
| Radial rescaling | $W_2 \sigma_{radial}(W_1 X)$ | $O(h)$ | $g \cdot (W_2, W_1) = (W_2 g^{-1}, g W_1)$ |
| Batchnorm | $\frac{(WX)_i - \mathrm{E}[(WX)_i]}{\sqrt{\mathrm{Var}[(WX)_i]}}$ | $\mathbb{R}_{>0}^h$ | $g \cdot W = g W$ |
| Softmax | $\mathrm{softmax}(WX)$ | $(\mathbb{R}^n, +)$ | $(g \cdot W)_i = W_i + g$ |
| Pointwise | $W_2 \sigma_{pointwise}(W_1 X)$ | $S_h$ | $g \cdot (W_2, W_1) = (W_2 g^{-1}, g W_1)$ |
| Radial basis function | $\sum_{i=1}^k w_i \varphi\left(\frac{\|\mathbf{x} - \mathbf{c_i}\|}{b_i}\right)$ | $S_k$ | $\pi \cdot (w_i, b_i, c_i) = (w_{\pi^{-1}(i)}, b_{\pi^{-1}(i)}, c_{\pi^{-1}(i)})$ |

**Example 2.10** (Attention (Vaswani et al., 2017))**.** *The self attention function is a core component of transformers. For key $K \in \mathbb{R}^{m \times p}$, query $Q \in \mathbb{R}^{n \times p}$, and value $V \in \mathbb{R}^{n \times r}$,*

$$Attention(QW^Q, KW^K, VW^V) = softmax\left(\frac{QW^Q(KW^K)^T}{\sqrt{d_k}}\right) VW^V$$

*with input embeddings $(Q, K, V) \in \mathbb{R}^{d \times d_m} \times \mathbb{R}^{d \times d_m} \times \mathbb{R}^{d \times d_m}$ and weights $(W_Q, W_K, W_V) \in \Theta = \mathbb{R}^{d_m \times d_k} \times \mathbb{R}^{d_m \times d_k} \times \mathbb{R}^{d_m \times d_v}$. This architecture has a $GL_{d_k}(\mathbb{R})$ symmetry, acting on $\Theta$ by $g \cdot (W^Q, W^K, W^V) = (W^Q g^{-1}, W^K g^T, W^V)$, for $g \in GL_{d_k}(\mathbb{R})$.*

Building on Example 2.10, multi-head attention replicates the attention block across $h$ heads and linearly mixes their outputs, thereby inheriting the $(\mathrm{GL}_{d_k}(\mathbb{R}))^h$ symmetry and introducing additional head-permutation ($S_h$) and per-head linear symmetries (($\mathrm{GL}_{d_v}(\mathbb{R}))^h$).

**Example 2.11** (Multi-head attention (Vaswani et al., 2017))**.** *Most transformer architectures use multi-head attentions, which concatenate the output of multiple attentions and apply a linear transformation to the concatenated output. Concretely, a multi-head attention is given by $MultiHead(Q, K, V) = Concat(head_1, ..., head_h)W^O$, with $head_i = Attention(QW_i^Q, KW_i^K, VW_i^V)$ and additional parameters $W^O \in \mathbb{R}^{hd_v \times d_m}$. The new parameter space is $\Theta = (\mathbb{R}^{d_m \times d_k} \times \mathbb{R}^{d_m \times d_k} \times \mathbb{R}^{d_m \times d_v})^h \times \mathbb{R}^{hd_v \times d_m}$. To simplify notation when describing symmetries, denote $W_i^O \in \mathbb{R}^{d_v \times d_m}$ as the matrix formed by row $(d_v \times i + 1)$ to row $(d_v \times (i + 1))$ of $W^O$.*

*In addition to the $(\mathrm{GL}_{d_k}(\mathbb{R}))^h$ symmetry inherited from individual attention heads, a multi-head attention admits an $S_h$ and a $(\mathrm{GL}_{d_v}(\mathbb{R}))^h$ symmetry. The $S_h$ symmetry acts as permutation of attention heads: $\pi \cdot (W_i^Q, W_i^K, W_i^V, W_i^O) = (W_{\pi^{-1}(i)}^Q, W_{\pi^{-1}(i)}^K, W_{\pi^{-1}(i)}^V, W_{\pi^{-1}(i)}^O)$, for $\pi \in S_h$. The multiplication of each attention head with $W_O$ results in one $\mathrm{GL}_{d_v}(\mathbb{R})$ symmetry, acting by $g_i \cdot (W_i^Q, W_i^K, W_i^V, W_i^O) = (W_i^Q, W_i^K, W_i^V g_i^{-1}, g_i W_i^O)$, for $g_i \in \mathrm{GL}_{d_v}(\mathbb{R})$, $i = 1, ..., h$.*

Parameters in a transformer can be finetuned after training to adapt to new datasets or tasks. A commonly used method, low-rank adaptation (LoRA) (Hu et al., 2022), updates pretrained parameters by low-rank matrices to reduce memory and compute. This parametrization adopts a general linear symmetry group.

**Example 2.12** (LoRA)**.** *In a low-rank adaptation $W + UV$, with pretrained weight matrix $W \in \mathbb{R}^{n \times m}$ and trainable parameters $U \in \mathbb{R}^{n \times r}, V \in \mathbb{R}^{r \times m}$, there is a $GL_r(\mathbb{R})$ symmetry, acting on $(U, V)$ by $g \cdot (U, V) = (Ug^{-1}, gV)$, for $g \in GL_r(\mathbb{R})$ (Putterman et al., 2024).*

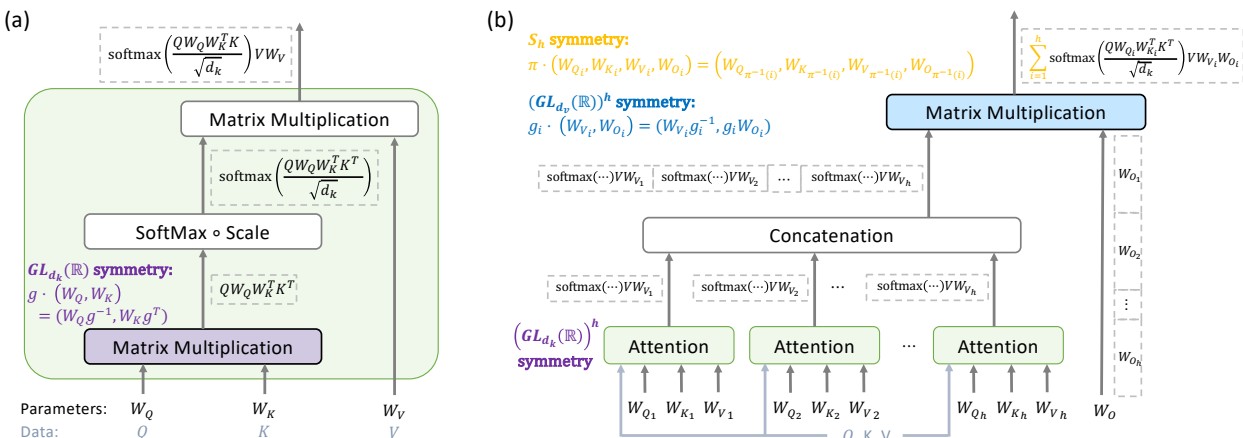

Figure 4: Symmetries in (a) the attention mechanism and (b) a multi-headed attention. Dashed rectangles represent output of each layer. Each symmetry is annotated in the same color as that of the component that gives rise to it.

## 2.3 Relaxed Definitions of Symmetry

While functional neural network symmetries ensure invariance of the neural network output across all input data, more general notions of symmetry arise when we relax this requirement. In this section, we expand the scope of parameter space symmetry to include transformations that preserve the overall loss function instead of the neural network function, as well as transformations that preserve the output over subsets of data instead of the entire data space.

### 2.3.1 Loss Symmetry

Loss symmetry refers to parameter transformations that preserve the overall loss but not necessarily the neural network function. Recall that the loss function $L\colon \Theta \times \mathcal{D} \to \mathbb{R}$ is often the composition of a model $f\colon \Theta \times \mathcal{D}_{\text{input}} \to \mathcal{D}_{\text{target}}$ and a cost function $c\colon \mathcal{D}_{\text{target}} \times \mathcal{D}_{\text{target}} \to \mathbb{R}$. Definition 2.4 defines transformations on $\Theta$ that do not alter the value of $f$, thereby also preserving $L$. We now relax this definition to include transformations that are required to preserve $L$ but allowed to change $f$.

**Definition 2.6** (Functional loss symmetry). *Let $\Theta$ be the parameter space of a loss function $L\colon \Theta \times \mathcal{D} \to \mathbb{R}$. A parameter space symmetry of $L$ is an action of a group $G$ on $\Theta$ that leaves $L$ invariant:*

$$L(g \cdot \theta, x) = L(\theta, x), \quad \forall g \in G, \quad \forall \theta \in \Theta, \quad \forall x \in \mathcal{D}.$$

**Example 2.13** (Self-supervised learning with linear network (Ziyin et al., 2023b)). *Consider a self supervised learning setting, with a linear network $f(W, x) = Wx$ and a loss $L(W, x)$ that depends on $f$ only through $f(x)^T f(x')$ for data pairs $x, x' \in \mathbb{R}^n$. Then, $L$ has a rotational symmetry, which acts on $\Theta = \mathbb{R}^{m \times n}$ by $g \cdot W = gW$ for $g$ in $O(m)$. This differs from the group action on linear networks (Example 2.2) since it only acts on one layer and is not canceled by transformations in other layers. Hence, this group action preserves the overall loss $L$ but not the feedforward function.*

### 2.3.2 Data-Dependent Symmetry

In the next definition, we relax the definition of functional symmetries (Definitions 2.4 and 2.6) by considering parameter transformations that preserve the output for data batches of size $n$, without guaranteeing invariance on other data. These symmetries depend on the data with respect to which the neural network output is invariant.

**Definition 2.7** (Data-dependent group action). *A data-dependent group action is a group action of $G$ on $(\Theta \times \mathcal{D}^n)$ that acts trivially on $\mathcal{D}^n$. To simplify notation, we drop $\mathcal{D}^n$ from the output and write a data-*

*dependent group action as a map $G \times (\Theta \times \mathcal{D}^n) \to \Theta$ that satisfies $e \cdot (\theta, X) = \theta$ and $g \cdot (g' \cdot (\theta, X), X) = (gg') \cdot (\theta, X)$ for all $g, g'$ in $G$, $\theta$ in $\Theta$, and $X \in \mathcal{D}^n$.*

**Definition 2.8** (Data-dependent symmetry). *Let $\Theta$ be the parameter space. Let $F \colon \Theta \times \mathcal{D} \to Y$ be a function with output space $Y$. A data-dependent symmetry of $F$ is a data-dependent action of a group $G$ on $\Theta$ that preserves the value of $f$ on a subset of data:*

$$F(g \cdot (\theta, X), x) = F(\theta, x), \quad \forall g \in G, \ \forall \theta \in \Theta, \ \forall X \in (\mathcal{D}_{input})^n, \ and \ \forall x \in X.$$

**Example 2.14** (Two-layer network (Zhao et al., 2023)). *For a nonzero vector $z \in \mathbb{R}^h$, define a matrix $R$ as*

$$(R_z)_{ij} = \begin{cases} z_i \cos(\alpha_{j-1}) \left( \prod_{k=1}^{j-1} \sin(\alpha_k) \right)^{-1} & \text{if } j \leq i \text{ and } \prod_{k=1}^{i-1} \sin(\alpha_k) \neq 0 \\ -r \sin(\alpha_i) & \text{if } j = i + 1 \\ 0 & \text{otherwise.} \end{cases}$$

*Consider a two-layer neural network $f(W_2, W_1, X) = W_2 \sigma(W_1 X)$, where $(W_2, W_1) \in \Theta = \mathbb{R}^{m \times h} \times \mathbb{R}^{h \times n}$ and $X \in \mathbb{R}^n$. Suppose $\sigma(z)$ is nonzero for any $z \in \mathbb{R}^h$. Then this architecture has a data-dependent $GL_h(\mathbb{R})$ symmetry, which acts on $\Theta$ by $g \cdot (W_2, W_1, x) = (W_2 R_{\sigma(W_1 x)} R_{\sigma(gW_1 x)}^{-1}, \ gW_1)$ and preserves loss value on single data points.*

### 2.3.3 Distribution Symmetry

In practical settings, data can often be viewed as samples from an underlying distribution. The following definition considers parameter transformations that preserve the expected loss over this distribution. Neural networks related by these transformations are expected to perform similarly on data within the given distribution but not guaranteed to perform similarly on data from different distributions.

**Definition 2.9** (Distribution symmetry). *Consider a function $F \colon \Theta \times \mathcal{D} \to Y$ with output space $Y$, where $\Theta$ is the parameter space and data are drawn from a distribution $D$. A distribution symmetry of $F$ is an action of a group $G$ on $\Theta$ that leaves the expectation of $F$ invariant:*

$$\mathbb{E}_{x \sim D}[F(g \cdot \theta, x)] = \mathbb{E}_{x \sim D}[F(\theta, x)], \quad \forall g \in G, \quad \forall \theta \in \Theta.$$

Distribution symmetry is often related to averaging effects, where aggregated predictions remain consistent despite variations in the predictions from individual parameters.

**Example 2.15.** *Consider the function $F \colon \mathbb{R}^{m \times n} \times (\mathbb{R}^n \times \mathbb{R}^n) \to \mathbb{R}^m$ defined by $F(W, (x, y)) = Wx - y$, where $W$ is the parameter and $x, y$ are data. Assume that $x$ is drawn from a distribution $D_x$ and for each data pair $(x, y)$, $y$ is defined as $y = W^* x$ for a fixed matrix $W^* \in \mathbb{R}^{m \times n}$. Let $\bar{x} = \mathbb{E}_{x \sim D_x}[x]$. Then, the expectation of $F$ under the data distribution is:*

$$\mathbb{E}_{x \sim D_x}[F(W, (x, y))] = \mathbb{E}_{x \sim D_x}[Wx - W^* x] = W\bar{x} - W^*\bar{x}.$$

*Let $G$ be the group of all invertible matrices $A$ for which $A\bar{x} = \bar{x}$. This group acts on the parameter space $\mathbb{R}^{m \times n}$ via the action $A \cdot W = WA^{-1}$. This action is a distribution symmetry for $F$, because for all $A \in G$ and matrix $W \in \mathbb{R}^{m \times n}$, we have:*

$$\mathbb{E}_{x \sim D_x}[F(A \cdot W, (x, y))] = \mathbb{E}_{x \sim D_x}[WA^{-1}x - W^* x] = W\bar{x} - W^*\bar{x} = \mathbb{E}_{x \sim D_x}[F(W, (x, y))].$$

The relation among the above definitions are visualized in Figure 5. Parameter transformations that preserve the neural network function are guaranteed to preserve the loss function. Parameter transformations that preserve a function at every data point are guaranteed to preserve the function over every fixed-size batch or distribution. Therefore, neural network symmetry (Definition 2.4) implies loss symmetry (Definition 2.6). Neural network (or loss) symmetry implies both neural network (or loss) data-dependent symmetry (Definition 2.8) and neural network (or loss) distribution loss symmetry (Definition 2.9).

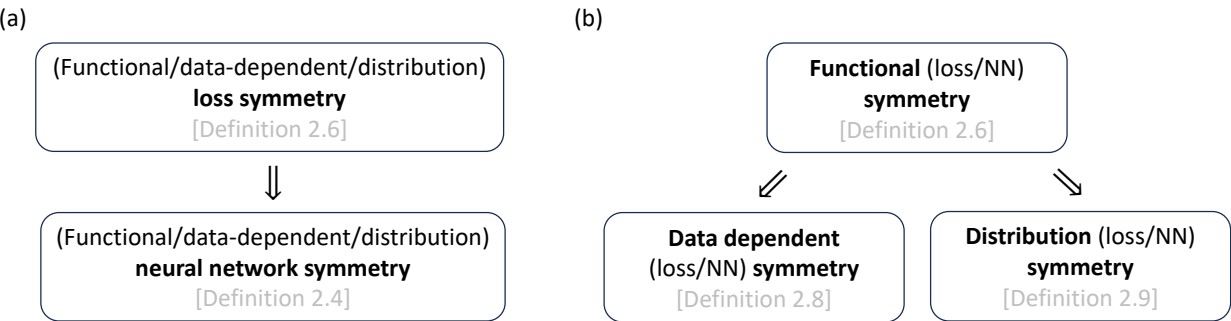

Figure 5: Relation among different definitions of parameter space symmetry. Arrows represent the relationship that satisfying one definition implies satisfying the other. (a) Symmetries that preserve the overall loss versus the neural network output. (b) Symmetries defined over the entire input space versus those defined over subsets of data.

## 2.4 Parameter Identifiability and Completeness of Symmetry

Having introduced a range of examples highlighting that symmetry is ubiquitous in neural network parameter spaces, we now turn to the fundamental question of whether known parameter symmetries account for all the ways in which parameters can represent the same function. This question concerns parameter identifiability—the extent to which a neural network's function determines its parameters, which is important in interpreting and comparing trained networks (Ran & Hu, 2017). This section formalizes the relationship between symmetry and identifiability using the realization map from parameters to functions, and surveys results on when known symmetries are sufficient to characterize all such redundancy.

Formally, we consider the realization map $\rho\colon \Theta \to \mathcal{F}$, where $\mathcal{F} = \{f\colon \mathcal{D}_{\text{input}} \to \mathcal{D}_{\text{output}}\}$, which associates each parameter $\theta \in \Theta$ with the function $\rho(\theta) \in \mathcal{F}$ it defines (Grigsby et al., 2025). The fiber of a function $f$ under $\rho$, denoted $\rho^{-1}(f) := \{\theta \in \Theta \mid \rho(\theta) = f\}$, is the set of all parameters that map to the same function $f$. In many architectures, $\rho$ is not injective, and fibers contain multiple distinct parameters that define the same function. This implies that parameters are, in general, not uniquely determined by the function alone.

Parameter space symmetries contribute directly to this non-identifiability, as they induce transformations within the same fiber. Therefore, analyzing fiber structures helps in determining whether the known set of symmetries is complete. If a symmetry group $G$ acts on $\Theta$ such that every element of a fiber can be reached from another via a transformation in $G$, we say $G$ acts transitively on that fiber. In this case, $G$ captures all the redundancy in the parameterization, and no hidden symmetries remain.

**Definition 2.10** (Parameter Identifiability). *Let $L\colon \Theta \times \mathcal{D} \to \mathbb{R}$ be a loss function with symmetry group $G$. We say parameters of $L$ are identifiable up to $G$ if for every fiber of the realization map of $L$, the group $G$ acts transitively on that fiber. We say $G$ is complete if parameters are identifiable up to $G$.*

For several architectures, such completeness has been established. For example, Gaussian radial basis function networks are known to be identifiable up to permutation symmetry, indicating that permutation symmetry is complete for these architectures (Kůrková & Neruda, 1994). For feedforward networks with tanh activations, if two networks represent the same function, then they have the same architecture and parameters up to permutation and sign-flip symmetries, under a known set of conditions (Sussmann, 1992; Fefferman & Markel, 1993). Similar completeness results have also been established for complex-valued tanh networks (Nitta, 2003; Kobayashi, 2010), tanh recurrent neural networks (Kimura, 2002), networks with activation functions $\sigma$ satisfying $\sigma(0) = 0, \sigma'(0) \neq 0$, and $\sigma''(0) = 0$ (Albertini & Sontag, 1993), and networks with asymptotically constant activation functions (Kůrková & Kainen, 1994). For Mixture-of-Experts (MoE) architectures, permutation-translation symmetries of the gating parameters were recently proven to fully characterize functional equivalence (Tran et al., 2025b).

The identifiability of ReLU networks is less straightforward. In particular, ReLU networks can contain hidden symmetries—transformations that preserve the realized function but not captured by permutation and positive scaling symmetries. Grigsby et al. (2023) describe several mechanisms through which hidden symmetries can arise. For example, for a single ReLU neuron with scalar input $x$ and realization map $\rho\colon (a,b) \mapsto (x \mapsto \mathrm{ReLU}(ax+b))$, the fiber of the constant function 0, $\rho^{-1}(0)$, contains $(0,0)$ and $(0,-1)$, which are not related by rescaling and permutation (Grigsby et al., 2025). To formalize this variability, Grigsby et al. (2025) introduce the functional dimension, defined as the rank of the Jacobian of the realization map $\rho$, which reflects the local dimensionality of the set of functions realized near a given parameter. They show that the functional dimension may not be constant even within the same fiber. This suggest a complicated structure of symmetries in ReLU networks.

Other recent work investigates when ReLU networks are identifiable up to positive scaling and permutation symmetries. (Petzka et al., 2020) prove that excluding a degenerate case where two neurons have identical zero hyperplanes, two-layer ReLU networks do not have additional symmetries besides permutation and positive scaling. (Rolnick & Kording, 2020) prove that under the assumption that networks within each linear region compute the same linear function, boundaries between linear regions uniquely determine parameters up to permutation and positive scaling. For ReLU networks with either non-increasing widths (Bui Thi Mai & Lampert, 2020) or all layers wider than input (Grigsby et al., 2023), a positive measure subset of the parameters have no symmetries besides permutation and positive scaling (hidden symmetries). Grigsby et al. (2023) also empirically show that the probability of having no hidden symmetry increases with input and layer width and decreases with depth. (Bona-Pellissier et al., 2023) provide sufficient conditions under which two ReLU network are identical up to permutation and positive scaling if their output agrees on a subset of the input space.

The identifiability of neural networks that are polynomial in their parameters can often be analyzed using tools from algebraic geometry (Marchetti et al., 2025). A notable example is multi-layer perceptrons with polynomial activation functions, or polynomial neural networks. Kileel et al. (2019) conjecture that permutation and scaling are the only symmetries in generic polynomial neural networks, which was later proved for the monomial case (Finkel et al., 2025). For more general polynomial activation functions, there are only finitely many data-independent parameter symmetries Shahverdi et al. (2025). Convolutional neural networks with polynomial activations generically have no nontrivial data-independent parameter symmetries. Another example of a polynomial function is lightening self-attention $(W^Q, W^K, W^V) \mapsto QW^Q(KW^K)^T V W^V$, where the softmax normalization is omitted from the standard attention mechanism (Example 2.10). Henry et al. (2025) prove that in networks composed of lightning self- attention layers, the only symmetries are (1) scaling $W_Q W_K^T$ and $W_V$ by a constant, (2) scaling $W_Q$ and $W_K$ by an invertible matrix, and (3) scaling $W_V$ of one layer and $W_Q, W_K$ in the next layer by an invertible matrix. They conjecture and verify numerically that adding back the softmax breaks only the first of these symmetries.

## 3  Role of Symmetry in Loss Landscapes

The loss landscape—the graph of the loss function over the parameter space—plays a central role in understanding the optimization behavior of neural networks. Parameter symmetries and the lack of parameter identifiability increase the size of loss level sets. To illustrate, consider a two–layer linear network $f_{linear}(W_2, W_1) = W_2 W_1 X$, where $(W_2, W_1) \in \Theta = \mathbb{R}^{m \times h} \times \mathbb{R}^{h \times n}$ and $X \in \mathbb{R}^{n \times k}$ is fixed input data. If $(W_2^*, W_1^*)$ is a global minimum, so are all points $(W_2^* g^{-1}, g W_1^*)$ with $g \in \mathrm{GL}_h(\mathbb{R})$. This set forms a positive-dimensional manifold of zero loss (Figure 6).

In this section, we examine how parameter space symmetry governs the connectedness, dimensionality, and structure of loss level sets. In Sections 4 and 5, we will see how these properties can be exploited in optimization.

### 3.1  Continuous Symmetry and Mode Connectivity

Continuous parameter space symmetries enlarge individual minima into connected sets. Formally, a symmetry is continuous if the symmetry group $G$ is a Lie group and the group action is continuous. The orbit

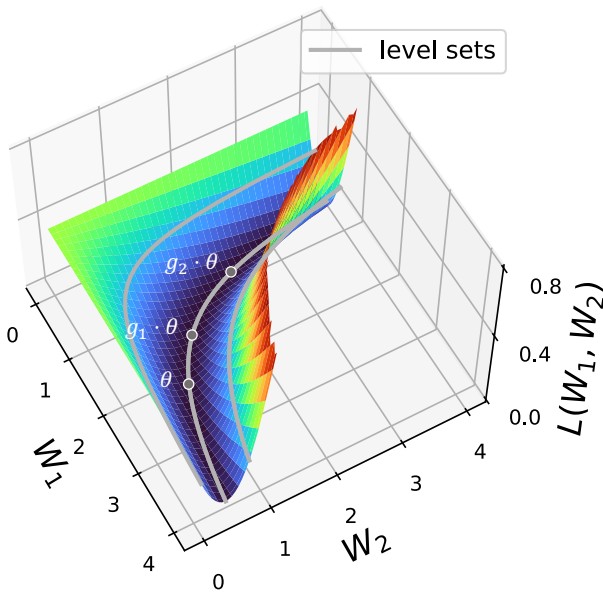

Figure 6: Parameter symmetries increase the size of the zero loss set. If $\theta$ is a global minimum, so are all points $g \cdot \theta$ with $g$ in the symmetry group of $L$. This set is typically positive-dimensional.

of a point $\theta \in \Theta$ is the set $O(\theta) = \{\theta' \in \Theta \mid \theta' = g \cdot \theta \text{ for some } g \in G\}$. When $G$ is a Lie group, the orbits are often positive-dimensional, so continuous symmetries generically create a positive-dimensional manifold lying entirely in a single loss level set.

Empirical work first revealed that independently trained networks can be joined by low-loss curves in parameter space (Garipov et al., 2018; Draxler et al., 2018). This observation, often referred to as mode connectivity (Garipov et al., 2018), has been extended to show that points found by stochastic gradient descent (SGD) are connected by multi-dimensional volumes (Benton et al., 2021). This insight has applications in model ensembling (Garipov et al., 2018; Benton et al., 2021; Benzing et al., 2022) and model averaging (Izmailov et al., 2018; Wortsman et al., 2022). Continuous symmetries provide a theoretical explanation for this phenomenon. The loss remains constant along each orbit, so minima discovered by SGD will automatically be connected whenever the group action is continuous and they can be mapped to each other by an element in the identity component of the group. We note that not all points in a minimum are related by a symmetry transformation, and empirical connectivity observed between independently trained networks may also arise from other mechanisms beyond symmetry.

Parameter space symmetry imposes a structure on the loss level sets. The minimum of a neural network, which is the optimization target, is perhaps the most interesting level set. Investigating the connection between symmetry groups and the topology of minima may lead to better understanding of the loss landscape and explain the source of mode connectivity. When a level set is homeomorphic to the symmetry group through a group action, e.g. in linear networks with invertible weight matrices, the minima has the same topological properties, such as connectedness, as the symmetry group (Zhao et al., 2025). Zero-loss curves generated by continuous actions (Zhao et al., 2024b) and dimension bounds derived from orbit dimensions (Zhao et al., 2023) further illustrate how symmetry dictates flat directions. (Lengyel et al., 2020) visualizes the set of functionally equivalent parameters. In ReLU networks, the visualization demonstrates the expected permutation and scaling symmetry, as well as other structures which indicate more complex symmetries.

### 3.2 Discrete Symmetry and Structure of Minimum

Discrete symmetries do not connect minima continuously; instead they replicate them, creating many functionally identical copies throughout the parameter space. Permutation is the best-studied discrete symmetry.

Brea et al. (2019) show that all permutations of a given hidden layer reside in the same loss level set, while Şimşek et al. (2021) prove that adding just one extra neuron per layer in a minimal tanh network merges these replicas into a single connected manifold. Extending this analysis, Farrugia-Roberts (2023) characterize functional equivalence classes for hyperbolic-tangent networks, and Pittorino et al. (2022) examine minima in the function space after quotienting out both permutation and scaling symmetries. Counting arguments reveal the combinatorial explosion such symmetries induce. Michelucci (2022) estimate that the number of permutation-equivalent minima grows factorially with width, underscoring how discrete symmetry complicates landscape exploration.

Although discrete actions do not generate continuous valleys, they still influence the loss landscape shape and optimization trajectories, as the next section will demonstrate. In particular, quotienting out these symmetries reduces model redundancy, shrinks the search space, and sometimes simplify sampling.

### 3.3 Removing Symmetry: Model Compression and Reduced Search Space

While symmetry enriches the loss landscape by introducing structure and multiplicity, removing symmetry reveals a simpler geometry—one that is more compact to represent, and often more amenable to analysis. Symmetry removal effectively collapses each symmetry orbit to a canonical representative (Sorensen, 2020). This operation alters the shape of the loss landscape—not by changing its values, but by restricting attention to a lower-dimensional quotient space where each point represents a unique function. As a result, the loss landscape becomes less redundant, more compact, and easier to sample from and optimize.

One major motivation for removing symmetry is to reduce the dimensionality of the parameter space without altering the function space. This operation, often known as model compression, is of critical importance in lowering storage cost, improving inference speed, and efficient deployment of machine learning models. Every symmetry implies a set of directions in which the loss is constant, stretching level sets into high-dimensional manifolds. Quotienting out these directions yields a more compact version of the loss landscape, where each point corresponds to a unique function. In the context of model compression, this means we can represent networks more efficiently. For example, Ganev et al. (2022) compress radial neural networks by factoring out orthogonal symmetries, and Sourek et al. (2021) exploit structural symmetries in computational graphs to merge redundant nodes. These methods reduce storage costs and inference time, without affecting expressivity or accuracy.

Beyond compression, removing symmetry transforms the loss landscape by collapsing symmetry-related regions into single representatives. For networks with permutation symmetry—such as feedforward networks with interchangeable hidden units—the parameter space can be partitioned into large equivalence classes. Building on the observation that permutations of neurons can be viewed as compositions of reflections in the parameter space, Hecht-Nielsen (1990) constructs a cone in the parameter space that contains a permuted copy of every point in the parameter space. For two layer networks with permutation symmetry $S_h$, the cone occupies only $1/h$ of the parameter space. They further prove the existence of a minimal search set, in which no two points are related by symmetry. This geometric reduction reduces the volume of minima and reveals its global structure. Lim et al. (2024b) empirically show that after removing permutation and scaling symmetries, neural networks are more linearly mode connected, and the loss decreases more monotonically on the linear interpolation between initialization and trained parameters.

In Bayesian neural networks, removing symmetry reduces the effective search space by collapsing functionally equivalent modes in the posterior (Lim et al., 2024b). In high-dimensional models like neural networks, symmetries can cause posterior distributions to become multi-modal in a functionally redundant way. This impairs MCMC mixing and complicates interpretation. By mapping parameters to a canonical representative—effectively quotienting out permutation or sign-flip symmetries—Wiese et al. (2023) and Laurent et al. (2024) show that it is possible to explore a more informative and diverse set of functionally distinct solutions. (Xiao et al., 2023) show that removing permutation symmetry leads to a compact representation that helps directly compare trained Bayesian neural networks across sampling methods.

Finally, removing symmetry alters the curvature and critical point structure of the landscape, with direct implications for optimization. Symmetry-induced flat directions often give rise to plateaus or saddle points that slow down training. By projecting the loss landscape onto a symmetry-reduced space, these degenerate

directions are removed, yielding a more strongly convex surface. Leake & Vishnoi (2021) show that continuous symmetries can be used to construct convex polytopes, allowing nonconvex optimization problems to be reformulated as linear optimization problems over polytopes, which are computationally more tractable.

In summary, removing symmetry simplifies the geometry and topology of the loss landscape, by collapsing orbits of equivalent parameters. This simplification yields a lower-dimensional loss surface, a more meaningful probabilistic structure in Bayesian neural networks, and an optimization landscape with fewer spurious critical points and degenerate directions. While symmetry endows the loss landscape with structure and multiplicity, its removal distills the parameter space into a representation more aligned with the underlying function space.

## 4 Applications of Symmetry in Gradient-Based Optimization

In this section, we describe ways of leveraging parameter space symmetry to design more efficient and theoretically motivated optimization algorithms. Optimization in machine learning is the process of finding parameters that minimize an objective function $L \colon \Theta \times \mathcal{D} \to \mathbb{R}$, over a given subset of data $X \subseteq \mathcal{D}$. In the absence of additional constraints on the parameters, the optimization problem is formulated as

$$\min_{\theta \in \Theta} L(\theta, X).$$

A common optimization method is gradient descent, which iteratively updates parameters in the direction opposite to the gradient $\nabla L(\theta) = \frac{\partial L}{\partial \theta}$. Using a step size $\eta_t \in \mathbb{R}_{>0}$, the parameter after $t+1$ steps is obtained from the parameter after $t$ steps by

$$\theta_{t+1} = \theta_t - \eta_t \nabla L(\theta_t). \tag{1}$$

When analyzing gradient descent, the presence of parameter symmetry introduces complexities that are not immediately apparent from the loss values alone. Consider two points in the parameter space, $\theta$ and $\theta' = g \cdot \theta$ for some $g$ in the symmetry group $G$. While the value of $L$ at $\theta$ and $\theta'$ is the same, the gradient $\nabla L$ may differ (Figure 7a). Therefore, learning dynamics starting from points in the same orbit can also be different (Van Laarhoven, 2017; Tanaka & Kunin, 2021) (Figure 7b).

The discrepancy among points in an orbit inspires two families of optimization techniques–purposefully searching for a favorable point within the orbit, or eliminating the differences among points in the same orbit. We discuss the two approaches in the next two subsections.

### 4.1 Exploiting Difference among Points in an Orbit

Leveraging the difference among points in an orbit, loss-invariant transformations defined by symmetry may be used in concert with common optimization algorithms to accelerate training or improve the quality of the final solution. During optimization, the parameter $\theta$ can be transformed by an element $g$ of the symmetry group $G$, $\theta \mapsto g \cdot \theta$ (Figure 7c). We can thus optimize over the orbits by optimizing over the group $G$. If $G$ is a Lie group, then optimization can likewise be performed using gradient descent on the manifold underlying $G$. Below we discuss various optimization objectives for $g$, including minimizing parameter norms, maximizing loss gradients, and optimizing other qualities of the solution (Table 2).

One application is minimizing the norm of the parameters, which has been shown to improve learning. On ReLU networks, Stock et al. (2019) and Saul (2023) search in the positive rescaling group for an element that minimizes parameter norms. Stock et al. (2019) propose Equi-normalization (ENorm), an algorithm that rescales parameters $\theta$ to minimize their $l_p$-norm $l_p(\theta) = \|\theta\|_p^p$. Saul (2023) proposes to minimize the $l_{p,q}$-norm $\|W\|_{p,q} = (\Sigma_i (\Sigma_j |(g \cdot W)_{ij}|^p)^{\frac{q}{p}})^{\frac{1}{q}}$, where $W_{ij}$ denotes the weight from neurons $i$ to $j$. In the resulting network, the incoming and outgoing weights are balanced at each layer. Interleaving these symmetry transformations with stochastic gradient descent improves training speed and test accuracy, following the intuition that gradient propagation is smoother on parameters with smaller norms.

Other work seeks to improve convergence speed by optimizing on the orbits to increase the norm of gradients. On general neural network architectures, Armenta et al. (2023) propose neural teleportation, which moves

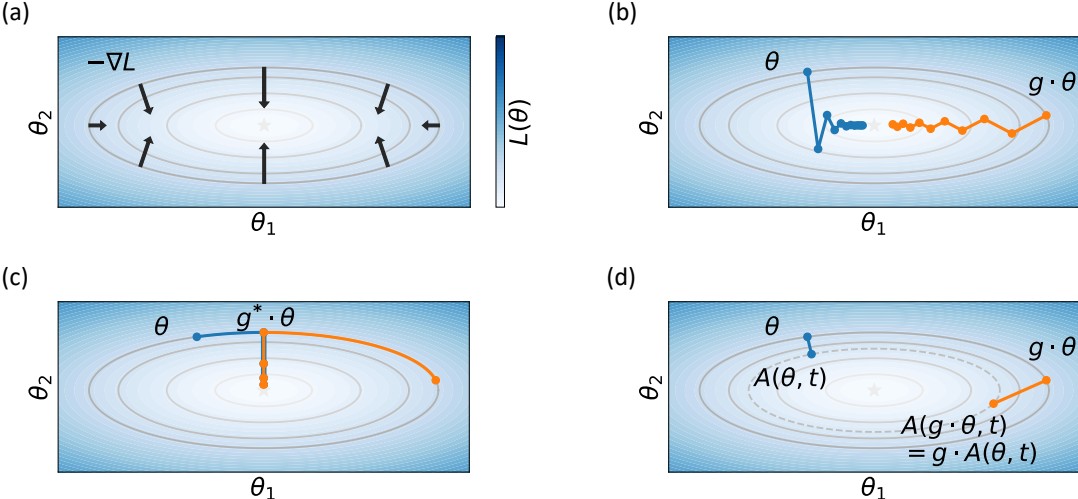

Figure 7: Implications and applications of parameter space symmetry in gradient-based optimization. (a) Points in an orbit have same loss values $L$ but different gradients $\nabla L$. (b) Gradient descent trajectories originating from two points in the same orbit, $\theta$ and $g \cdot \theta$, may differ. (c) Algorithms leveraging these differences search within the orbit for a point that leads to faster optimization ($g^* \cdot \theta$). (d) A $G$-invariant algorithm $A$, which commutes with the group action, is indifferent to initializations within an orbit.

parameters on their loss level set using a group element random drawn from a distribution on the symmetry group. This procedure leads to an increase in the expected magnitude of gradients, resulting in improved empirical convergence rates. Building on this work, Zhao et al. (2022) perform optimization in the orbit to find points with large gradient norms $\|\nabla L\|_2$. They show that points where gradient norms are maximized in a loss level set, the gradient descent direction $\nabla L$ aligns with Newton's direction $H^{-1} \nabla L$, where $H$ is the Hessian of $L$. This alignment suggests that optimal teleportation before each gradient descent step enables the algorithm to achieve the faster convergence rates typical of second-order methods. In preconditioned gradient descent $\theta_{t+1} = \theta_t - \eta A \nabla L(\theta_t)$, where $A$ is a matrix with dimensions matching $\Theta$, teleporting to maximize the Mahalanobis norm $\|\nabla L\|_A$ results in faster convergence compared to maximizing the $l_2$-norm $\|\nabla L\|_2$ (Zhao et al., 2024b). Since its introduction, teleportation has found broad application in long-tail recognition (Yang et al., 2025), continual optimization in multi-task learning (Zhou et al., 2025), and privacy-preserving verification of deep learning inference (Maheri et al., 2025).

Beyond finding a good starting point for gradient descent, optimization on orbits can incorporate other objectives at different times during training. For example, Zhao et al. (2024b) teleport parameters to regions with different sharpness and curvature, empirically improving the generalization ability of the final solution. More generally, knowing the symmetries allows one to search on the minima for points with desired properties. When the optimization on orbits is done using infinitesimal symmetries, this approach is similar to orthogonal gradient descent in continual learning (Farajtabar et al., 2020), which projects the gradient of new tasks to the space orthogonal to the gradient of the original task to avoid catastrophic forgetting.

Group actions that preserve loss approximately but not exactly are still a useful tool to explore near a loss level set. In this case, one can leverage the different loss or functions realized by points in the same orbit. In certain methods with regularizers, loss is not invariant on the orbits, but the variation is small, which slows down optimization after parameters enter the orbit. This is analogous to the Goldstone mode in physics (Altland & Simons, 2010). After optimization on the full parameter space slows down due to entering the Goldstone mode, Bamler & Mandt (2018) propose to optimize on the symmetry subspace, which accelerates optimization in these orbits with weakly-broken symmetry.

Table 2: Algorithms utilizing group elements to transform parameters during optimization.

| Algorithm | Reference | Group Element for Parameter Update |
|---|---|---|
| ENorm | Stock et al. (2019) | $\arg\min_{g \in G} \|g \cdot \theta\|_p^p$ |
| Weight Balancing | Saul (2023) | $\arg\min_{g \in G} \left( \Sigma_i \left( \Sigma_j |(g \cdot W)_{ij}|^p \right)^{\frac{q}{p}} \right)^{\frac{1}{q}}$ |
| Teleportation | Armenta et al. (2023) Zhao et al. (2022) Zhao et al. (2024b) | sampled from a distribution on $G$ $\arg\max_{g \in G} \|(\nabla L)\|_{g \cdot \theta}\|_2^2$ $\arg\max_{g \in G} \text{Curvature}(g \cdot \theta)$ |
| Goldstone-GD | Bamler & Mandt (2018) | $\arg\min_{g \in G} L(g \cdot \theta)$ |

## 4.2 Symmetry Invariant Optimization Algorithms

Points in a symmetry group orbit share the same loss but can differ in other aspects. In Section 4.1, we explored optimization algorithms that capitalize on these differences. A complementary approach of handling these differences involves designing learning algorithms that remain unaffected by such differences. In this section, we discuss ways to make gradient descent invariant to parameter symmetries.

Similar to symmetry in the loss function, symmetry in algorithms is defined by the invariance of an algorithm to group actions on parameters. We denote an algorithm as $A \colon \Theta \times \mathcal{D} \times \mathbb{R} \to \Theta$, which takes initial parameters with a dataset and outputs the learned parameters after $t$ steps. Inspired by the definition of data-equivariant learning algorithms (Abbe & Boix-Adsera, 2022), the following definition generalizes rescaling invariant algorithms (Neyshabur et al., 2015) to general parameter-equivariant algorithms, visualized in Figure 7d.

**Definition 4.1** (*G*-invariant algorithm). *Let $G$ be a symmetry group of a loss function $L \colon \Theta \times \mathcal{D} \to \mathbb{R}$. An algorithm $A$ is $G$-invariant if, for every $g \in G$, $\theta \in \Theta$, $X \subseteq \mathcal{D}$, and $t > 0$, there exists an $g' \in G$ such that $A(g \cdot \theta, X, t) = g' \cdot A(\theta, X, t)$.*

### 4.2.1 Scaling Invariant Algorithms

ReLU networks are positively rescale-invariant (Example 2.3), but gradient descent (Equation 1) on them is not. When parameters are rescaled in ways such that there are significant disparities between input and output weights of neurons, gradient descent tends to perform poorly (Neyshabur et al., 2015). The extra effort required to circumvent these configurations, along with difficulty in analysis caused by inconsistent performance across parameters on the same orbit, motivates new optimization algorithms that are invariant to rescaling.

Two families of rescale invariant algorithms have been developed. The first one is based on the observation that the product of incoming and outgoing parameters of a ReLU function is invariant to rescaling. From the parameters $\theta$ of a ReLU network, Neyshabur et al. (2015) construct a path vector $\pi(\theta)$ that contains the product of parameters along each path from an input node to an output node. Their algorithm, Path-SGD, performs proximal gradient descent with respect to a path regularizer $\|\pi(\theta) - \pi(\theta^t)\|_p^2$, instead of the 2-norm of parameters $\|\theta - \theta^t\|_2^2$ that would give rise to Equation 1:

$$\theta^{t+1} = \arg\min_\theta \eta \left\langle \nabla L(\theta^t), \theta \right\rangle + \frac{1}{2} \left\| \pi(\theta) - \pi(\theta^t) \right\|_p^2.$$

The resulting update step size for each parameter is inversely proportional to the norm of a vector, where each entry is the product of all other parameters along a path that includes the parameter. Consequently, parameters with larger values receive proportionally larger updates, and the resulting algorithm achieves rescaling invariance.

Another method to make gradient descent invariant to scaling symmetry involves projecting parameters to a different space, then either optimizing directly in that space or, when the space is not a vector space,

performing manifold optimization. The space is often a quotient space induced by scaling equivalence and has fewer dimensions than $\Theta$. Meng et al. (2019) design a rescaling invariant algorithm, $\mathcal{G}$-SGD, by performing gradient descent in the rescale-invariant space spanned by the path vector of basis paths. Badrinarayanan et al. (2015) and Huang et al. (2020) constrain the incoming weights of each neuron to stay unit-norm by updating the weight matrix $W$ of each layer using the Riemannian gradient before projecting them back to the oblique manifold $\{W \in \mathbb{R}^{n \times p} : \text{diag}(WW^T) = I\}$. In addition to ReLU networks, other architectures such as those with batch normalization also exhibit scaling symmetry (Example 2.6), and similar optimization techniques apply. Yi (2022) quotients out the positive scaling symmetry in batch normalization, develops gradient descent algorithms on the quotient manifold in a similar way, and proves that the resulting algorithm has a better convergence rate than gradient descent in the original parameter space.

### 4.2.2 General Symmetry Invariant Algorithms

One algorithm whose convergence is invariant to general parameter symmetry is natural gradient descent. This method computes gradients using the Fisher metric on the manifold of distributions, which is invariant to parametrizations (Amari, 1998). In implementation, natural gradient descent is not fully invariant to symmetry due to finite step size. However, methods which use a second-order ODE solver and the second-order approximation of the exponential map can reduce the invariance error to second order (Song et al., 2018). More recently, Kristiadi et al. (2023) observe that gradient descent can be made invariant to parameter symmetry by explicitly including the metric when computing gradients, and show that there exist metrics other than the Fisher metric that achieve this invariance. Scale-free updates, which are updates invariant to gradient scaling, are hypothesized to be the reason why AdamW outperforms standard Adam with $\ell_2$ regularization (Zhuang et al., 2022).

## 5 Implications of Symmetry for Learning Dynamics

In this section we continue to explore how parameter space symmetry informs learning dynamics. In physical systems, Noether's theorem states that certain continuous symmetries have associated conservation laws (Noether, 1918). Similarly, in neural network training, parameter space symmetry implies the existence of conserved quantities in gradient flows. These conserved quantities, which remain constant along gradient flow trajectories, are useful both as conditions held throughout training and as a way to parameterize optimization trajectories in the parameter space. We summarize known conserved quantities arising from parameter space symmetry and discuss their roles in theoretical analysis of learning dynamics.

### 5.1 From Symmetry to Conserved Quantities

Gradient flow is a continuous version of gradient descent, commonly used to analyze learning dynamics in the limit of infinitesimal step sizes. It defines a trajectory $\theta(t) \in \Theta$ for $t \in \mathbb{R}_{>0}$ that connects an initialization to the limiting critical point, with velocity given by the gradient of loss: $\dot{\theta}(t) = -\nabla_{\theta(t)}L$. A conserved quantity of gradient flow is a function $Q \colon \Theta \to \mathbb{R}$ that remains constant along this trajectory, i.e., $Q(\theta(s)) = Q(\theta(t))$ for all $s, t \in \mathbb{R}_{\geq 0}$.

A widely studied conserved quantity in gradient flow is imbalance—the difference between the Gram matrices of adjacent layers in linear networks.

**Example 5.1** (Imbalance (Arora et al., 2018b))**.** *Consider an l-layer linear feedforward network $f \colon \Theta \times \mathcal{D}_{input} \to \mathcal{D}_{target}$, defined as $(W_1, ..., W_l), X \mapsto W_l...W_1 X$, with parameter space $\Theta = \mathbb{R}^{n_l \times n_{l-1}} \times \cdots \times \mathbb{R}^{n_1 \times n_0}$. Define the loss function $L \colon \Theta \times \mathcal{D} \to \mathbb{R}$ as the composition of $f$ and any differentiable function $c \colon \mathcal{D}_{target} \times \mathcal{D}_{target} \to \mathbb{R}$. For $h \in [l-1]$, each element in $W^{(h)}(W^{(h)})^T - (W^{(h+1)})^T W^{(h+1)} \in \mathbb{R}^{n_h \times n_h}$ is a conserved quantity of the gradient flow on L, known as the unbalancedness (Du et al., 2018) or imbalance (Tarmoun et al., 2021).*

Similar conserved quantities have been identified beyond linear feedforward networks and gradient flow. For example, in feedforward networks with homogeneous activations, Du et al. (2018) show that gradient flow preserves the difference in squared norms between each neuron's incoming and outgoing weights, making it a conserved quantity. In graph attention networks with homogeneous activation functions, Mustafa et al.

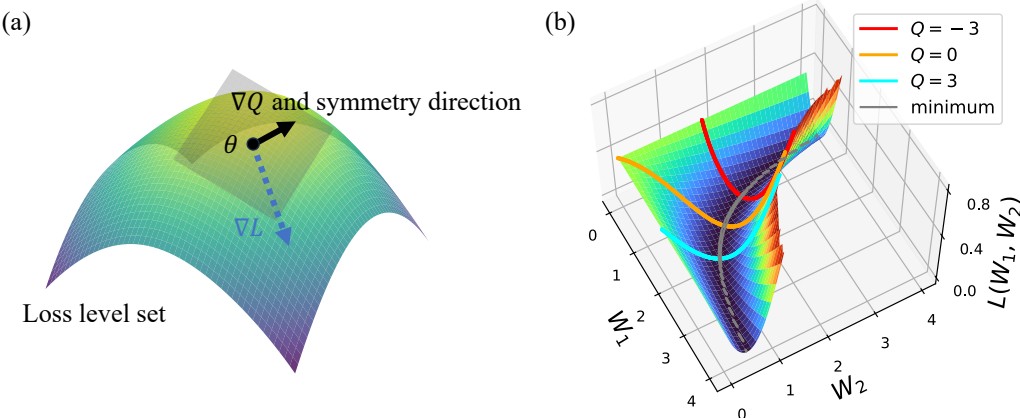

Figure 8: (a) Symmetry and corresponding conserved quantities. Symmetry directions–infinitesimal actions of parameter space symmetries–and the gradients of conserved quantities, $\nabla Q$, are both orthogonal to $\nabla L$ and hence lie tangent to the loss level set shown in the figure. Symmetries and conserved quantities can therefore be connected by matching $\nabla Q$ to a symmetry direction. (b) Conserved quantities partially parameterize gradient flow trajectories and minima (adapted from Figure 1 in Zhao et al. (2023)). Different trajectories correspond to distinct values of the conserved quantity $Q$, which remains fixed along the trajectory. The minima reached by these trajectories also have distinct conserved quantity values.

(2023) identify similar conserved quantities—the difference between the norm of incoming and outgoing parameters of each neuron. By expressing convolutional layers as equivalent fully connected layers, Le & Jegelka (2022) identify conserved quantities of similar forms between convolutional layers and between residual blocks in ResNet. Beyond standard gradient descent, Huh (2020) identify conserved quantities under spectral initialization in natural gradient descent and curvature-corrected dynamics, including expressions involving differences and ratios of singular values of weight matrices.

Several theoretical frameworks explicitly connect conserved quantities to parameter space symmetries, drawing analogies to Noether's theorem and extending conservation laws across diverse optimization dynamics. For example, Kunin et al. (2021) derive conserved quantities in gradient flows and analyze their evolution under modified gradient flows that better approximate stochastic gradient descent. Their work focuses on one-parameter symmetry groups, such as translation, scaling, and rescaling, and parallels Noether's theorem in both form and intuition. Extending these ideas to higher-dimensional symmetry groups, Głuch & Urbanke (2021) develop a framework for describing optimization algorithms as ordinary differential equations, identifying their corresponding Lagrangians, and deriving conserved quantities by applying Noether's theorem via the generators of symmetry groups. Notably, they connect imbalance to symmetry and derive conserved quantities for a broader class of dynamics, including both first-order and second-order systems such as Newtonian dynamics and Nesterov's accelerated flows.

For more general architectures, Zhao et al. (2023) associate conserved quantities with infinitesimal symmetries, deriving them for activation functions with general equivariance properties. This approach recovers known conserved quantities for homogeneous activations and reveals a new conservation law analogous to angular momentum conservation for radial rescaling activation layers.

While many conserved quantities can be derived from parameter space symmetries, it remains an open question whether all conserved quantities originate from known symmetries. To explore whether known sets of conserved quantities are maximal, Marcotte et al. (2023) apply the Frobenius theorem to compute the number of independent conserved quantities, from the Lie algebra generated by the vector fields spanned by the model's Jacobian. In the case of the matrix factorization problem (Example 2.2 without the bias terms), they show that besides imbalance, there are no other independent conservation laws. Similar results hold for attention layers in transformers (Marcotte et al., 2025).

## 5.2 Conserved Quantities for Convergence and Parameterization

By ensuring certain properties remain constant throughout training, conserved quantities provide stability in gradient flow that is essential for convergence analysis. In particular, any condition based solely on conserved quantities that holds at initialization will continue to hold throughout the gradient flow. As noted in (Głuch & Urbanke, 2021), conserved quantities such as imbalance provide guarantees that parameters are bounded throughout gradient flow, which is helpful in convergence analysis. The proofs of several convergence bounds for gradient descent of deep linear networks depend either on the invariance of the imbalance (Ji & Telgarsky, 2019) or the assumption that the imbalance is small or zero (Arora et al., 2018a;b). Zero imbalance is also an assumption in (Bah et al., 2022), as a condition under which the gradient flows of linear networks are Riemannian gradient flows on the manifold of fixed rank matrices.

Beyond stability, conserved quantities act as coordinates for the dynamics. Because they are constant along a trajectory, they label optimization paths, and their limiting values locate endpoints within the set of minima. This labeling links initialization to both convergence and generalization: in small two-layer networks, Zhao et al. (2023) observe strong correlations between conserved quantities and both convergence and sharpness of the resulting minima, suggesting that carefully chosen initializations can improve training efficiency and generalization. Moreover, under imbalanced initializations, conserved quantities have been explicitly connected to the convergence rate by appearing as terms in convergence bounds (Tarmoun et al., 2021; Min et al., 2021; 2023; Xu et al., 2023).

By parametrizing minima, conserved quantities also help describe which solutions optimization converges to, formalizing the notion of implicit bias – the tendency of optimization algorithms to favor solutions with specific properties. In homogeneous and leaky ReLU networks, Du et al. (2018) and Kou et al. (2023) prove that layers become automatically balanced. Similarly, Wang et al. (2022) proves an implicit regularization effect in matrix factorization problems under large learning rates. This effect is quantified by an upper bound on the difference between the 2-norms of weight matrices, a quantity similar to imbalance. This trend also appears empirically–Kunin et al. (2021) observe that imbalance decreases exponentially when training with large learning rates. For loss functions with a rescaling symmetry, (Ziyin et al., 2023a) derive the stationary distribution of SGD and prove that SGD solutions are biased towards a balanced one, different from predictions from a Langevin model. Extending this idea, Ziyin (2024) propose a unified framework based on mirror reflection symmetry, encompassing rescaling, rotation, and permutation symmetries. They prove that such symmetries impose structured constraints, which are preferentially satisfied when weight decay or gradient noise is large. This framework explains various phenomena in gradient-based learning and motivates algorithms that enforce constraints.

As such, conserved quantities, rooted in symmetry, bridge the structure of parameter space and the dynamics of learning, guiding both the path and outcome of optimization.

# 6 Connections to Symmetry in Internal Representations and Data

In previous sections, we focused on how symmetry impacts the structure and dynamics within the parameter space. In this section, we explore how parameter space symmetry interacts with symmetry in other spaces, including data and internal representations. In the first part, we examine how activation function equivariance links symmetry in parameter space to symmetry in internal representations. We then discuss how symmetry in data can induce symmetry in learned parameters, followed by a brief look at an application involving joint transformations of data and parameters. In the second part, we turn to tasks where neural network parameters are treated as data, and show how parameter symmetry can be leveraged through equivariant architectures and data augmentation.

## 6.1 Symmetry in Parameters, Internal Representations, and Data

Symmetry in parameter space is closely connected to symmetry in internal representations, the intermediate outputs of neural networks, with both often arising from the equivariance of activation functions. In Section 2.2, we saw that many parameter space symmetries arise from equivariance of the activation functions

$\sigma$. Recall that $\sigma$ is equivariant to a group $G$ if it satisfies $\sigma \circ g = g \circ \sigma$ for $g \in G$. The set of such transformations, known as the intertwiner group, defines symmetries in both the parameter space and the internal representations of data (Godfrey et al., 2022). This property enables stitching: given two functionally equivalent networks with the same architecture but different weights, expressed as function compositions $f_1 \circ \sigma \circ f_2$ and $\tilde{f}_1 \circ \sigma \circ \tilde{f}_2$, an element $g \in G$ can align the two via $\tilde{f}_1 \circ g \circ \sigma \circ f_2$ while preserving the output.

Symmetries in internal representations also motivate new approaches for comparing hidden activations across networks. In particular, Godfrey et al. (2022) point out that internal representations should be considered equivalent up to transformations from the intertwiner group, leading to similarity metrics that are invariant under such transformations. This perspective justifies interpreting ReLU networks through the behavior of individual neurons, whose activations are preserved under these symmetries, rather than through arbitrary linear combinations, which may obscure meaningful structure.

In two-layer ReLU networks trained with gradient descent, symmetry in training data preserves symmetry in the learned parameters (Anselmi et al., 2023). Specifically, let $X \subseteq \mathcal{D}$ be a dataset that is invariant under a group $G$, i.e., $g \cdot X = X$ for all $g \in G$. This implies the loss function satisfies $L(\theta, g \cdot X) = L(\theta; X)$. Consider a two-layer ReLU network of the form $v^T \sigma(Wx)$, where $(v, W) \in \Theta = (R^k \times \mathbb{R}^{k \times d})$ and $x \in \mathcal{D} = \mathbb{R}^d$. Because of this data invariance, the gradient with respect to $W$ is equivariant under $G$, i.e. $\nabla_W L(g \cdot W; X) = g \cdot \nabla_W L(W; X)$. As a result, if $W$ is initialized to have the same symmetry as data ($gW_0 = W_0$ for all $g \in G$), this symmetry is preserved throughout training ($gW_t = W_t$ for all $g \in G$, at all time $t$).

While most studies examine symmetries acting on either the data space or the parameter space, it is also possible to define simultaneous transformations on both spaces that leave the loss function invariant. For example, in a one-layer linear network $f(W, b, X) = WX + b$, the group action $g \cdot (W, X) = Wg^{-1}, gX$ leaves the output unchanged. By formulating such joint group actions on data and parameters, Sonoda et al. (2025) reveal a group theoretic aspect of neural network approximation theory. Using Schur's lemma, they show that the ridgelet transform is a right inverse of the integral representation of neural networks. This means that the ridgelet transform can reconstruct a neural network that exactly represents a given data-generating function. As a result, their framework provides a constructive proof of the universal approximation theorem, rooted in the symmetry structure of the joint space.

## 6.2 Parameter Symmetry as Data Symmetry in Weight Space Learning

In tasks where neural network parameters are treated as data–for example, in parameter generation (Chauhan et al., 2024) or in learning representation of trained models (Schürholt et al., 2024b)–the parameter space symmetry of the input network becomes the data space symmetry of the processing network. Designing architectures that process neural network parameters effectively has attracted increasing attention (Schürholt et al., 2024a), partly due to the convergence of several trends: the rise of neural networks that directly encode data objects, such as implicit neural representations (Park et al., 2019); the need for weight alignment in model merging (Ainsworth et al., 2023; Wortsman et al., 2022); and growing interest in model analysis driven by the increasing amount of publicly released trained models (Horwitz et al., 2025). Advances in methods that process neural network parameters as data have enabled applications including learning on implicit neural representations (Dupont et al., 2022), optimizing weight alignment between networks (Navon et al., 2024), predicting generalization ability (Unterthiner et al., 2020), recovering the dataset size used in fine-tuning (Salama et al., 2024), and generating high-performing parameters (Abbe & Boix-Adsera, 2022; Wang et al., 2024) or high-quality initializations (Knyazev et al., 2023).

Respecting parameter space symmetry of the input network is a desirable property of these architectures, as it ensures consistent treatment of functionally equivalent networks and improves generalization and efficiency. This concept aligns with the broader goals of geometric deep learning–a field that leverages structures in data to design more effective learning methods (Gerken et al., 2023; Bronstein et al., 2021). The rest of this section reviews how these ideas are implemented through equivariant architectures and data augmentation.

One way to enforce data symmetry in processing architectures is to use equivariant neural networks (Cohen & Welling, 2016; Ravanbakhsh et al., 2017; Kondor & Trivedi, 2018; Maron et al., 2019) (Figure 9). Applying this idea to processing multi-layer perceptrons (MLPs), Navon et al. (2023) propose an architecture that is equivariant to permutation of the MLP's hidden neurons. Their architecture outperforms nonequivariant

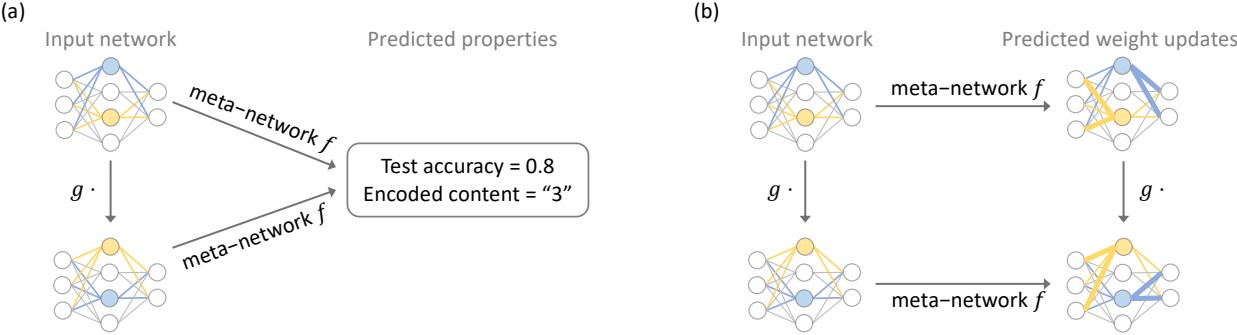

Figure 9: Invariant and equivariant metanetworks. (a) A metanetwork $f$ is invariant to $G$ if $f(g \cdot x) = f(x)$ for all input network $x$. Invariance is desirable in tasks such as predicting generalization performance or classifying INRs, where the output should remain unchanged under parameter symmetries. (b) A metanetwork $f$ is equivariant to a symmetry group $G$ if, for all input network $x$, $f(g \cdot x) = g \cdot f(x)$. This is used in tasks such as learning to optimize (predicting update steps) or style editing for INRs (predicting weight changes in INRs that produce a given style change in the encoded content).

models on implicit neural representation classification and prediction, as well as the adaptation of classification networks to new domains. Concurrently, Zhou et al. (2023a) develop a similar architecture that is equivariant to permutation of all neurons and input of convolutional neural networks. Subsequently, this architecture is extended to process arbitrary weight spaces (Zhou et al., 2024) and to account for scaling symmetry in ReLU (Tran et al., 2024), although the proposed models can only learn from inputs with one fixed architecture. Lim et al. (2024a) and Kofinas et al. (2024) treat neural networks as computational graphs and use graph neural networks to process the input neural networks. Their models are thus able to learn from diverse architectures. Most recently, Kalogeropoulos et al. (2024) extend graph-based metanetworks to account for scaling symmetries arising from activation functions, achieving state-of-the-art performance across multiple neural network processing tasks.

Building on the success of equivariant architectures that leverage parameter symmetry, later work applies them to a wider range of input networks and tasks. For example, Zhou et al. (2023b) propose transformers that are equivariant to permutation symmetry in the input networks, while Tran et al. (2025a) develop architectures that are equivariant to symmetry in transformers. Incorporating the knowledge of parameter symmetry into solving the weight alignment problem, Navon et al. (2024) extend the equivariant architecture in Navon et al. (2023) to learn the optimal alignment between two sets of parameters. Their approach is faster, produces better alignment than traditional optimization-based approaches, and can be used as initialization for optimization-based methods to improve their alignment quality.

Beyond equivariant networks, parameter space symmetry can also be leveraged through data augmentation. Applying symmetry transformations to existing trained parameters generates new, functionally equivalent versions of the network to be used as additional training inputs. While most existing equivariant architectures focus on permutation symmetry, Shamsian et al. (2024) introduce scaling symmetry in ReLU networks and discrete symmetries in sinusoidal activations to generate new sets of parameters as training samples. These augmentations increase the diversity of neural representations for each data object, mitigating overfitting in weight space learning and improving generalization.

## 7 Challenges and Future Directions

In Sections 2–6, we reviewed the current understanding of parameter space symmetries, covering their theoretical foundations as well as tangible effects on the geometry of loss landscapes, the dynamics of optimization, and the design of practical algorithms. In this section, we outline key research directions towards a more complete theoretical foundation of parameter space symmetry, better understanding of the role of symmetry in deep learning theory, and broader applicability of symmetry-informed methods.

### 7.1 Mathematical Foundations

A fundamental challenge in understanding parameter space symmetry is to develop a complete and rigorous characterization of all symmetries that preserve neural-network functions. From a mathematical perspective, clarifying the existence, completeness, and structures of these symmetry groups represents a foundational problem. Even seemingly simple network architectures can have nontrivial symmetry groups, and identifying them is essential for understanding loss landscapes and optimization dynamics.

Another challenge is extending the notion of symmetry from exact, function-preserving transformations to more flexible, data-dependent symmetries. We saw in Section 2.4 that in many neural networks with elementwise activation functions, all transformations that do not change the function are known. However, this set of symmetry is often small, as they are required to keep the loss invariant for all input values. Data-dependent symmetry allows for larger symmetry groups, but current analysis is limited to symmetry that preserves the loss for a single data point. Investigating the existence and structure of data-dependent symmetry for different batch sizes will to make this set of symmetry more relevant to practical settings.

Thus far, most work has focused on layer-wise equivariances arising from adjacent activations, potentially overlooking more global invariances. Existing studies examine small components—pairs of layers whose activations admit a group action—but this does not preclude broader classes of symmetry that act across multiple, non-adjacent layers or repeated architectural blocks. Exploring such symmetries could substantially expand the known symmetry groups and reveal new geometric or topological structures in parameter space.

Besides pursuing a general theory, detailed investigations into architecture-specific symmetries are also valuable. Models such as neural radiance fields (Mildenhall et al., 2021), which are computationally expensive to optimize, might benefit greatly from symmetry-informed optimization methods. Discovering and characterizing symmetries particular to these architectures would provide concrete opportunities to accelerate training and enhance scalability, thereby translating theoretical advances into immediate practical improvements.

Finally, numerical and visualization tools can help guide theoretical characterization of parameter space symmetries. Methods for numerically constructing and visualizing functionally equivalent parameters, such as those developed by Lengyel et al. (2020), provide practical insight into the connectedness, dimensionality, and topological complexity of symmetry-induced loss level sets. Numerical discovery of symmetries (Zhao et al., 2024a) and symmetry-induced structures (Martinelli et al., 2024) in the parameter space may also provide intuitions on the existence and number of symmetries in a given architecture. These computational approaches can be important in guiding theoretical efforts in large-scale architectures.

### 7.2 Deep Learning Theory

Parameter space symmetry is not merely a mathematical curiosity—it offers a lens to understand core phenomena in deep learning theory. As neural networks scale in size and complexity, symmetry provides a principled foundation for analyzing learning dynamics, the geometry of loss landscapes, and model capacity. Symmetry considerations are thus increasingly central in explaining how overparameterized networks behave, generalize, and learn in practice.

**Training dynamics and implicit bias.** One promising direction is to study learning dynamics through the lens of symmetry and conserved quantities. As we have seen in Section 5, continuous symmetries induce conserved quantities that remain fixed along gradient flows and partially parameterize the minimum. These quantities label the optimizer's position and reveal how initialization and symmetry constrain the final solution, formalizing implicit bias. However, the precise relationship between conserved quantities, implicit bias, and generalization remains poorly understood—especially in deeper architectures or when SGD breaks these symmetries due to finite-step updates. Understanding how stochasticity or regularization cause the drift of these conserved quantities will be an alternative route to explain phenomena in training neural networks, such as why stochastic gradient descent prefers minima that generalize.

**Loss landscape geometry and generalization.** Symmetry is an integral part of the geometry of the loss landscape, which is closely linked to generalization. By expanding minima into manifolds of equivalent

solutions, symmetries create flat directions that confound curvature-based metrics such as sharpness. Recent work by da Silva et al. (2025) addresses this issue by formulating sharpness on a Riemannian quotient manifold that factors out the continuous symmetry of transformers, recovering a strong correlation between curvature and generalization. Developing such symmetry-aware geometric formalisms may help reconcile conflicting observations about the relationship between flatness and generalization.

**Expressivity and approximation theory.** Symmetry has direct implications for the expressivity and effective capacity of neural networks. Since many parameter configurations can represent the same function (e.g., due to permutation symmetry in multilayer perceptrons), the function class is smaller than suggested by raw parameter counts. Recent work by Shen (2024) quantifies this, showing that factoring out symmetry leads to significantly tighter covering number bounds—improving estimates by factorial factors in width. These results suggest that traditional complexity measures overestimate model capacity by ignoring symmetric redundancies. Extending such analyses to broader symmetry groups (beyond permutations) could yield sharper generalization bounds by treating symmetry-related parameters as equivalent in the function space.

**Introducing or removing parameter symmetries?** Architectural design offers a means to either introduce or eliminate symmetries, which provides the opportunity to examine symmetry's effect on various aspect of deep learning. On one hand, imposing symmetry can enhance training stability and invariance. For example, Li et al. (2022) introduce a scale-invariant transformer, replacing Softmax with a normalized ReLU to create a model trainable by plain SGD yet competitive with Adam-trained BERT variants. On the other hand, removing symmetry reduces degeneracies, creates more connected minima, and sometimes makes optimization easier. Lim et al. (2024b) propose architectures that break permutation symmetry between neurons, enabling linear mode connectivity and improving Bayesian training efficiency. Additionally, (Ziyin et al., 2025) show that removing symmetries via small perturbations can prevent capacity collapse and improve both optimization and generalization by encouraging exploration of more expressive model configurations. An important open question is determining which symmetries to preserve and which to break, in order to best align model behavior with the goals of a given learning task.

Taken together, these directions position symmetry not only as a descriptive tool for neural networks, but as a foundational principle for understanding learning dynamics, designing architectures, and guiding future theory in deep learning.

## 7.3 Applications

**Optimization on loss level sets.** One practical use of symmetry is to facilitate movement along loss level sets in the parameter space. Points on a fiber of the realization map can have very different neighborhoods (Grigsby et al., 2025). In other words, points on the same level set can be different. Early work illustrates the usefulness of this property in optimization (Armenta et al., 2023). Despite initial promise, these methods have not been wide adapted, possibly due to a lack of large-scale benchmark study and an easy to use implementation, as well as the mathematical background required to adapt these algorithms to new architectures. New implementation techniques (Mishkin et al., 2023; Wu et al., 2025) and learning the teleportation destination (Zhao et al., 2024b; Zamir et al., 2025) might help reduce the cost of teleportation in larger scale applications.

Besides scaling up teleportation, another promising direction is to explore level sets via symmetry transformation in diverse domains. When some parts of the minima are better than others, one can use symmetry to explore the minima to find the better models. One example is continual learning, where one optimize in the minimum manifold (Farajtabar et al., 2020) and could be a method for fine-tuning pre-trained models. Another example is model alignment, where symmetry is used to move models closer in the parameter space to improve model fusion (Ainsworth et al., 2023; Zhang et al., 2025). Recent work also demonstrates that parameter space symmetries can be leveraged to align sparse subnetworks across different random initializations, enabling effective sparse training from scratch (Adnan et al., 2025). Moreover, some minima may produce lower error during quantization than others (Meller et al., 2019; Nagel et al., 2019; Laird et al., 2025). Given a known symmetry and a point on minimum, we can search for these better points on an orbit.

**Reducing search space in sampling.** Symmetry can be exploited beyond standard gradient-based training, for instance in Bayesian methods and sampling-based optimization (Wiese et al., 2023). The parameter space explored by sampling algorithms (such as MCMC for neural network weights) may also contain redundancies coming from symmetry, though this aspect remains less studied. Factoring out symmetries in such scenarios could reduce the effective search space and improve sampling efficiency. For example, if parameters related by a continuous symmetry produce the same likelihood, one could constrain the sampler to move only in the reduced space of unique configurations, thereby eliminating needless exploration of equivalent states. This idea can also be used as a diagnostic: known symmetries provide a consistency check for the quality of sampling (the sampler should visit all members of a symmetry orbit with equal probability). Discrete symmetry has been explored in depth (Wiese et al., 2023; Laurent et al., 2024; Xiao et al., 2023), but factoring out continuous symmetry might be more appealing, since it reduces the dimension, instead of just volume, of the posterior space.

## 8 Conclusion

Symmetry is prevalent in neural network parameter spaces and appears in many areas of machine learning, though often overlooked. Recognizing and formalizing these symmetries connects deep learning to well-established mathematical tools from group theory and geometry. Symmetry's intrinsic connection to structure improves our understanding of how neural networks work and hints at better architectures, faster optimization techniques, and new approaches to solving problems with real-world impact.

Symmetry is, of course, not the only path forward in the study of neural networks. Like modeling training dynamics using classical mechanics or designing new neural networks with inspirations from neuroscience, it is an example of approaching problems in machine learning from the view of a pre-existing subject, allowing us to bring existing tool and insight to bear on the problem. What makes a mathematical approach especially appealing is that neural networks are fully artificial, abstract objects, unconstrained by the specific laws of physics or biology. By examining parameter space symmetry, we gain access to powerful mathematical frameworks that help us better understand these systems and ultimately design more effective models.

## Acknowledgements

We thank Kathlén Kohn and Hancheng Min for helpful comments. This work was supported in part by the U.S. Army Research Office under Army-ECASE award W911NF-07-R-0003-03, the U.S. Department Of Energy, Office of Science, IARPA HAYSTAC Program, and NSF Grants #2205093, #2146343, #2134274, #2442658, #2134178, CDC-RFA-FT-23-0069, DARPA AIE FoundSci and DARPA YFA.

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
