# OpenReview forum: "Symmetry in Neural Network Parameter Spaces"
_TMLR — Accepted by TMLR_

### Review · Reviewer_yUTA · 2025-07-21

**Summary Of Contributions:**

This work is a **survey** (as there are **no original results**) on some results on parameter simmetries in neural networks. Just to give some concrete examples, the Authors go through the standard permutation symmetry (Example 2.8) and positive rescaling symmetry in homogeneous networks (Example 2.3) and other actions that leave the output of a neural network unchanged. The authors later extend the notion of symmetry in weaker settings (see Figure 5, with e.g. data / distribution symmetries defined). Later, the authors connect symmetries to the existence of connections between different minima of the loss landscape and compression, and to ideas to improve gradient based optimization, namely with techniques such as gradient teleportation and parameter regularization. Finally the authors mentions existing results on conserved quantities during gradient optimization.

**Audience:**

Yes

**Claims And Evidence:**

Yes

**Requested Changes:**

I would kindly ask the Authors if they could consider my comments, and do a pass to remove repetition, maybe compress the presentation, and remove the most vague statements. Repeating myself: to the best of my knowledge the optimization methods that non-trivially rely on the symmetry of the parameter space are not currently used in the most modern architectures. While this does not undermine the value of this submission, I do believe elaborating on this could make the presentation more factual, and therefore better.

**Strengths And Weaknesses:**

This paper is **carefully written**. The Authors clearly put a lot of work in reviewing existing literature, providing a neat presentation of the results, including well explanatory tables and figures. While the paper itself does not seem to bring any original contribution (neither theoretical or empirical) a work like this one can be useful for readers interested in the topic.

Regarding the weaknesses, I do think the paper gets (at times) a bit **verbose**, **rethorical**, and **speculative**, especially in the later sections when the authors switch from providing examples and definitions to elaborate more qualitatively on the roles of symmetry for the loss landscape, optimization, ... (starting  from Sections 3). Here are few examples

- End of page 10: not sure I am following the logic. Two modes can be connected even if there is no element in the symmetry group that maps one in the other.

- Section 3.3: model compression is, in general, done very differently than via the underlying symmetries of neural networks. While the authors mention 2 cases where symmetries are indeed considered, it is probably good to mention that this is not the current workhorse behind model compression.

- End of page 11: The Authors write "This geometric reduction _simplifies_ the landscape". This is a bit of a vague statement, and similar comments hold for later parts (e.g. "_easier_ to solve" at the end of the third paragraph at page 12. Also "more convex" can be better phrased: maybe add "strongly"?).

- Section 4.1: this is a very positive presentation of approaches that speed up optimization algorithms. However, to the best of my knowledge, these algorithms are not commonly used to train models at scale. Could the authors elaborate on why? Is the problem the different architectures or are these methods not giving any advantage in larger models (same question holds for Section 4.2.1)? Furthermore, the last paragraph in this section seems a bit sloppy to me: how is a "relaxed symmetry" defined? Maybe the Authors could spend just a few more words here.

- Section 5.2: is this Section truly necessary? It seems quite redundant with 5.1 and the part on optimization.

- Section 7: I would say also very verbose and with some repetitions / vague statement (e.g., "providing a concrete handle on implicit bias" in 4th paragraph page 21, "could help explore flatter or higher performing regions of the loss landscape, or improve convergence by starting in a favorable orbit" in the paragraph after, repetition on the teleportation algorithm in the first paragraph of Section 7.3)

---

> ### Author Response · Authors · 2025-09-20
>
> > End of page 10: not sure I am following the logic. Two modes can be connected even if there is no element in the symmetry group that maps one in the other.
>
> The original wording suggested that all empirically observed connectivity could be explained by continuous symmetry, which is indeed too strong. We edited the paragraph to clarify that symmetries guarantee connectivity along orbits, but connectivity can also arise from other mechanisms, even when no symmetry maps one solution to another.
>
> > Section 3.3: model compression is, in general, done very differently than via the underlying symmetries of neural networks. While the authors mention 2 cases where symmetries are indeed considered, it is probably good to mention that this is not the current workhorse behind model compression.
>
> We agree and added a note in the second paragraph of Section 3.3.
>
> > End of page 11: The Authors write "This geometric reduction simplifies the landscape". This is a bit of a vague statement, and similar comments hold for later parts (e.g. "easier to solve" at the end of the third paragraph at page 12. Also "more convex" can be better phrased: maybe add "strongly"?).
>
> We made the statements more precise accordingly. Specifically, we changed the phrase "simplifies the landscape" to “reduces the volume of minima”, and expanded “easier to solve” by explaining that symmetry allows certain nonconvex optimization problems to be reformulated as linear optimization problems over polytopes, which are computationally more tractable.
>
> > Section 4.1: this is a very positive presentation of approaches that speed up optimization algorithms. However, to the best of my knowledge, these algorithms are not commonly used to train models at scale. Could the authors elaborate on why? Is the problem the different architectures or are these methods not giving any advantage in larger models (same question holds for Section 4.2.1)? Furthermore, the last paragraph in this section seems a bit sloppy to me: how is a "relaxed symmetry" defined? Maybe the Authors could spend just a few more words here.
>
> Symmetry-enhanced optimization algorithms have indeed not been widely adopted. While the exact reason is unclear to us, we speculate that the lack of a large-scale benchmark study, the lack of a python package that is easy to use, and the mathematical background required to adapt these algorithms to new architectures all contribute to the barrier. We hope to see studies on the performance of these algorithms on larger models and have added a discussion on this point in future directions.
>
> We removed the phrase “relaxed symmetry”, which was not defined nor referred to in the previous manuscript. We explained the group actions that approximately preserve loss in the later part of that paragraph.
>
> > Section 5.2: is this Section truly necessary? It seems quite redundant with 5.1 and the part on optimization.
>
> We keep Section 5.2 because it serves a distinct purpose. Section 5.1 defines conserved quantities and shows how they arise from symmetry; Section 5.2 uses them to (i) obtain stability/convergence bounds, (ii) parameterize optimization trajectories, and (iii) parameterize the minima. While (ii) is related to the optimization method in Section 4.1, the focus here is analytical - deriving bounds using conserved quantities - rather than algorithmic. To address redundancy, we removed repeated sentences and tightened the section.
>
> > Section 7: I would say also very verbose and with some repetitions / vague statement (e.g., "providing a concrete handle on implicit bias" in 4th paragraph page 21, "could help explore flatter or higher performing regions of the loss landscape, or improve convergence by starting in a favorable orbit" in the paragraph after, repetition on the teleportation algorithm in the first paragraph of Section 7.3)
>
> We appreciate the suggestions and edited these statements accordingly. We also shortened this section by removing other repetition of ideas and tuned down verbosity.

---

### Review · Reviewer_uQ9x · 2025-08-22

**Summary Of Contributions:**

This manuscript thoroughly surveys parameter space symmetry by (i) reviewing definitions and examples of symmetries in neural network parameter space; (ii) analyzing the applications of parameter space symmetry; and (iii) providing related open questions and research directions.

**Audience:**

Yes

**Claims And Evidence:**

Yes

**Requested Changes:**

1. As noted in Weakness 1, the authors should consider adding analysis in Sections 2–6 when reviewing existing work. For instance, Section 4.2 currently reads as a plain description of methods; a higher-level comparison would make this section more valuable.
2. As noted in Weakness 2, the authors are encouraged to include small illustrative experiments. Even minimal empirical evidence could greatly help readers connect the theoretical discussions to practical outcomes.
3. Regarding Figure 4, the color scheme could be improved for clarity. Specifically, the reviewer found the blue $GL_{d_v}$ symmetry confusing—does it correspond to the matrix multiplication shown on the right? The color coding appears inconsistent, and clearer labeling or adjustments would help avoid ambiguity.

**Strengths And Weaknesses:**

Strength:
1. The paper is well written, with clear structure and illustrations. The examples (e.g., Figure 2) help the readers to better understand the abstract concept.
2. The paper is comprehensive in scope, which covers the parameter space symmetry in relation to loss, optimization and learning algorithms, which makes the survey comprehensive.

Weakness:
1. As a survey paper, the original theoretical or empirical contributions is limited. Section 2-6 primarily review the existing work, but do not consistently emphasize or synthesize new, previously unreported connections between these works.
2. The paper is highly theoretical with minimal experimental evidence, which may limit its accessibility to a broader audience and reduce its practical impact. Even small illustrative experiments or synthetic case studies could strengthen the presentation.

---

> ### Author Response · Authors · 2025-09-20
>
> > As a survey paper, the original theoretical or empirical contributions is limited. Section 2-6 primarily review the existing work, but do not consistently emphasize or synthesize new, previously unreported connections between these works.
>
> We respectfully disagree that our contributions are limited to simply summarizing existing work. As this is a survey, we do not introduce new theorems or experiments, but we do provide a unified perspective for parameter space symmetry that incorporates various innovations in architectures, optimization, learning theory, and representation learning. Specifically:
> - **Framework-level contributions:** We define multiple types of symmetry (functional, loss, data-dependent, distributional) and formalize them via group actions and representations (Section 2.3; Figure 5), providing precise mathematical structure to what has often been discussed informally.
> - **Unifying theme:** We connect work on optimization trajectories, conserved quantities, mode connectivity, implicit bias, and generalization under the common lens of symmetry. To our knowledge, this is the first survey that makes explicit connections across such a broad range of previously siloed results (Sections 3–5).
> - **Original taxonomy and synthesis:** Our structuring of the literature—for instance, distinguishing between exploiting symmetry differences (Section 4.1) and designing invariant algorithms (Section 4.2)—offers new conceptual clarity. Similarly, in Section 6, we draw formal parallels between parameter symmetry and symmetry in data and internal representations, bridging two distinct research areas.
>
> It’s important to note that many of the works we cite viewed their contributions as highly disparate from each other or relevant to certain specific areas such as optimization or model alignment.  The study of parameter space symmetry as a distinct area is relatively new and it’s a valuable contribution to recontextualize these older works within this overall framework (which mathematically, they absolutely belong to.)
>
> > Section 4.2 currently reads as a plain description of methods; a higher-level comparison would make this section more valuable.
>
> We believe Section 4.2 provides more than a plain description of methods. We introduce a formal definition of $G$-invariant algorithms (Definition 4.1), organize methods into two main symmetry classes - scaling and general parameter symmetries, and connects these methods to theoretical motivations (e.g., convergence stability, function-level equivalence) and empirical performance issues (e.g., imbalance in ReLU networks). We have adjusted the wording to further clarify the differences between algorithm families.
>
> > The paper is highly theoretical with minimal experimental evidence, which may limit its accessibility to a broader audience and reduce its practical impact. Even small illustrative experiments or synthetic case studies could strengthen the presentation.
>
> We appreciate the suggestion. As a survey paper, our primary goal is to synthesize and unify existing work on parameter space symmetry, not to introduce new experimental results. That said, we have made a deliberate effort to enhance accessibility by including concrete examples to ground abstract definitions (Table 1), figures, and clear practical implications in Sections 4-6, including symmetry-aware optimization, model alignment, and weight-space learning. We believe the theoretical clarity, unified framework, and breadth of coverage are of greater value in this case, and look forward to future papers that focus on systematic empirical studies.
>
> > Regarding Figure 4, the color scheme could be improved for clarity. Specifically, the reviewer found the blue symmetry confusing—does it correspond to the matrix multiplication shown on the right? The color coding appears inconsistent, and clearer labeling or adjustments would help avoid ambiguity.
>
> Yes, the blue symmetry corresponds to the matrix multiplication shown on the right. We have adjusted the color of the matrix multiplication box to match the symmetry description text more closely.

---

### Review · Reviewer_iTx9 · 2025-09-09

**Summary Of Contributions:**

This survey paper focuses on providing a overview of parameter space symmetry. It summarizes existing theoretical and algorithmic applications, emphasizing the connections to learning theory, optimization, and model analysis. In addition, the pape discusses how parameter space symmetry interacts with data and representation symmetries, offering a unified view of their role in shaping network behavior and performance.

**Audience:**

Yes

**Broader Impact Concerns:**

No.

**Claims And Evidence:**

Yes

**Requested Changes:**

The paper would be better to compare with the asymmetry property in some neural networks, providing a comprehensive perspective of he position of symmetry in neural networks. Perhaps involving a subsection to discuss would be better.

**Strengths And Weaknesses:**

Strengths:

1. The paper is well-written and easy to follow.

2. The paper covers diverse analysis on parameter space symmetry, including theory (group theory, geometry, and topology), optimization, conserved quantities, and interactions with data and representation symmetries, thus providing a clear understanding of parameter space symmetry.

Weaknesses:

1. Some expressions in the paper lack depth. For example, using “Symmetry is prevalent in deep learning architectures” as the first sentence of Section 2 looks too generic. It should benefit from a more substantial introduction.

2. Some figures in the paper do not look professional. For example, in Figure 8(a), the placement and font of the legend or notation are not professional.

3. The opening future research questions proposed in Section 7 are not creative and novel. They appear more like direct extensions or specific descriptions of the perspectives already covered in Sections 3-6.

---

> ### Author Response · Authors · 2025-09-20
>
> > Some expressions in the paper lack depth. For example, using “Symmetry is prevalent in deep learning architectures” as the first sentence of Section 2 looks too generic. It should benefit from a more substantial introduction.
>
> Upon rereading the manuscript, we agree that it is premature to make this statement before listing definitions and examples. We thus remove the sentence and leave a more grounded summary in the introduction paragraph of Section 2.
>
> > Some figures in the paper do not look professional. For example, in Figure 8(a), the placement and font of the legend or notation are not professional.
>
> We changed the font in Figure 8(a) to match the font of the text and adjusted the position of notations. We appreciate the suggestion and are happy to make further adjustments as needed.
>
> > The opening future research questions proposed in Section 7 are not creative and novel. They appear more like direct extensions or specific descriptions of the perspectives already covered in Sections 3-6.
>
> We intended Section 7 to propose future directions that are well-grounded in the themes developed in Section 2-6 and to identify concrete and actionable research questions. We tightened Section 7 to highlight how the series of research questions unify the field of parameter space symmetry.
>
> > The paper would be better to compare with the asymmetry property in some neural networks, providing a comprehensive perspective of the position of symmetry in neural networks. Perhaps involving a subsection to discuss would be better.
>
> Thank you for the suggestion. We included a paragraph at the end of Section 7.2 that discusses removing symmetry in neural networks and its effect on linear mode connectivity, Bayesian neural network training, and optimization in general.

---

### Author Response · Authors · 2025-09-20
**Summary of Changes**

We thank all reviewers for the helpful feedback and have improved the manuscript accordingly. We removed repetitions and vague statements, shortened Section 7, elaborated on limitations of existing methods, and updated Figures 4 and 8. We also added discussions on symmetries in polynomial neural networks and LoRAs, which came to our attention during the review period.

---

### Decision · Action_Editor_mB68 · 2025-10-29

**Recommendation:** Accept as is

**Additional Comments:**

This survey of symmetries in neural network parameter spaces provides a valuable reference for researchers and those interested in the topic. It is well-written and appears comprehensive, touching on training dynamics, optimization, expressivity, geometry, as well as various theoretical and practical considerations that derive from these topics. The reviewers raised some minor comments that were addressed in the rebuttal. Overall, this provides a solid contribution to TMLR and I recommend acceptance as-is.

**Audience:**

Yes

**Audience Explanation:**

Yes, the topic is salient and the community will have interest in this paper

**Claims And Evidence:**

Yes

**Claims Explanation:**

Yes, the claims are supported by evidence